# BOOSTING TEMPORAL GRAPH LEARNING FROM GLOBAL AND LOCAL PERSPECTIVES

## ABSTRACT

Extensive research has been dedicated to learning on temporal graphs due to its wide range of applications. Some works intuitively merge GNNs and RNNs to capture structural and temporal information, while recent works propose to aggregate information from neighbor nodes in local subgraphs based on message passing or random walk. These methods produce node embeddings from a global or local perspective and ignore the complementarity between them, thus facing limitations in capturing complex and entangled dynamic patterns when applied to diverse datasets or evaluated by more challenging evaluation protocols. To address the challenges, we propose the **G**lobal and **L**ocal **E**mbedding **N**etwork (GLEN) for effective and efficient temporal graph representation learning. Specifically, GLEN dynamically generates embeddings for graph nodes by considering both global and local perspectives. Then, global and local embeddings are elegantly combined by a cross-perspective fusion module to extract high-order semantic relations in graphs. We evaluate GLEN on multiple real-world datasets and apply several negative sampling strategies. Sufficient experimental results demonstrate that GLEN outperforms other baselines in both link prediction and dynamic node classification tasks.

## 1 INTRODUCTION

Graph representation learning (Hamilton et al., 2017b; Battaglia et al., 2018) has attracted tremendous research interest in both academic (Perozzi et al., 2014; Tang et al., 2015) and industrial (Wang et al., 2018; Rossi et al., 2019) communities owing to its powerful capabilities of mining and discovering abundant information in the non-Euclidean space (Asif et al., 2021; Wu et al., 2020). However, general methods consider the graphs to be static. Nothing is eternal except change itself. In the real world, most graph systems are usually dynamic and constantly change over time, making temporal graphs ubiquitous (Longa et al., 2023; Ma et al., 2020). In such temporal graphs, the topologies of networks evolve as nodes and edges appear or disappear across different timestamps, along with the attributes of nodes and edges changing dynamically (Zhu et al., 2022; Du et al., 2018). Learning on temporal graphs has received substantial research attention (Kazemi et al., 2020) since the ability to process dynamic networks can be useful for a wider range of scenarios like recommender systems (Wang et al., 2021a; Zhang et al., 2021), biology and medicine (Loo et al., 2023; Lim et al., 2019), traffic forecasting (Zhao et al., 2019), pandemic forecasting (Panagopoulos et al., 2021), etc.

There has been a surge of solutions for temporal graph learning (Souza et al., 2022; Wang et al., 2021c; Rossi et al., 2020). Many works sophisticatedly combine graph neural networks (GNNs) (Kipf & Welling, 2016; Velickovic et al., 2017) and recurrent neural networks (RNNs) (Medsker & Jain, 1999) to obtain structural and temporal information. These GNN-RNN methods take the entire graph at each time step as the input of GNNs and dynamically update the weight parameters of GNNs (Pareja et al., 2020; Chen & Hao, 2023) or the node features (Liu et al., 2020; Chen et al., 2022) through RNNs. Whereas temporal graph networks (TGNs) (Souza et al., 2022) generate dynamic node representations through aggregating temporal subgraphs triggered by events. Such approaches utilize memory modules with message passing mechanism (MP-TGNs) (Xu et al., 2020; Rossi et al., 2020) or aggregate temporal walks (WA-TGNs) (Wang et al., 2021c; Bastas et al., 2019b). GNN-RNN methods model temporal graphs from a global perspective, but lack the information of micro variation. TGNs obtain the features of each node by aggregating information from a limited neighboring region without perceptions of global structural dependencies. The unidimen-

| Temporal graph learning methods | | Preservation of neighborhood information when generating node embeddings | | | Modeling of temporal information | | | | Integrity of graph structure | |
|---|---|---|---|---|---|---|---|---|---|---|
| | | | | | Form | | Granularity | | | |
| | | Enables all edges to be utilized | Possible to discard noisy edges | Possible to discard useful edges | RNNs | Time Encoding | Coarse | Fine | Global graph | Local subgraph |
| **Global Perspective** | GNN-RNN methods | ✓ | | | ✓ | | ✓ | | ✓ | |
| **Local Perspective** | MP-TGNs | | ✓ | ✓ | | ✓ | | ✓ | | ✓ |
| | WA-TGNs | | ✓ | ✓ | | ✓ | | ✓ | | ✓ |

Figure 1: In the temporal graph example (a), edges $e_{AB}$ and $e_{AC}$ occur at $t_1$, while $e_{CD}$ occurs at $t_2$. Different methods (b) and (c) produce node embeddings in different ways and have different properties. Node embeddings are used to capture the correlation between nodes, so as to make predictions (e.g., whether nodes $B$ and $D$ will interact at $t_3$).

sionality of the aforementioned methods could result in less accurate inferences (Lu et al., 2019). In this paper, we present that modeling temporal graphs from both global and local perspectives is advantageous (Jin et al., 2019) in the following aspects.

**The neighborhood information gathered in two ways has certain complementarities.** Different methods retain or discard different neighborhood information. As shown in Figure 1, GNN-RNN methods (Pareja et al., 2020; Liu et al., 2020; Manessi et al., 2020) retain all events of each time step without filtering, so that all edges are fully utilized when generating node embeddings, but noisy or useless edges are also retained. As for TGNs methods, the neighborhood size is generally limited by a given constant via sampling operations (Rossi et al., 2020; Wang et al., 2021c; Zheng et al., 2021). Sampling of the temporal neighbors provides the opportunity to avoid noisy and irrelevant edges but may cause some interactions to be ignored or futilely reused when updating the states of nodes.

**The temporal information acquired from two perspectives complements each other.** The diversity of graph topologies across different domains leads to the complexity of temporal properties. Due to the regularity and abruptness of events, the pattern of events can also vary across time. Therefore, modeling at different time granularities have to be taken into account. RNNs learn the evolution patterns between adjacent graph snapshots at a coarse level. In contrast, MP-TGNs and WA-TGNs encode timestamps simultaneously while aggregating neighborhood contextual information (Xu et al., 2020; Rossi et al., 2020; Wang et al., 2021c). These two types of approaches model the temporal relevance of event occurrence in different forms and with different granularities as indicated in Figure 1, thus the acquired temporal information can complement each other.

**The two types of methods retain different integrities of graph structures.** Since the endogenous and exogenous factors driving the generative process of networks are frequently complex and variable, temporal graphs across diverse domains tend to exhibit a variety of properties (Zheng et al., 2021). For instance, social networks and international trade networks may have extremely different characteristics (e.g., varying sparsities and edge recurrence patterns) (Poursafaei et al., 2022). GNN-RNN methods with the global perspective are more likely to consider the overall nature of a temporal graph since GNNs maintain the complete graph structure at different time steps. In contrast, fine-grained patterns in motifs (Paranjape et al., 2017; Liu et al., 2021) such as the triadic closure process (Zhou et al., 2018; Liu et al., 2022) are better reflected in the encoding of local subgraphs by TGNs.

Based on the aforementioned insights, we propose GLEN [1] (short for ***Global and Local Embedding Network***) to learn representations for temporal graphs by considering both global and local perspectives. Our method fills the research gap in existing temporal graph methods that only focus on one perspective and highlights the importance of considering both perspectives. Unlike conventional global-view methods that model sequences using RNNs, we employ a temporal convolution network (TCN) for more efficient and stable training. From the local perspective, we devise a weighted sum algorithm based on time interval to distinguish the impact of events at different time. Since neither GNN-RNN methods, MP-TGNs, nor WA-TGNs can extract high-order features in graphs (Mao et al., 2023; Xu et al., 2018; Talati et al., 2021), simply fusing embeddings of two perspectives via summation or concatenation is empirically less than ideal. To tackle this issue, we devise a cross-perspective fusion module for GLEN to combine the node features embedded from global and local perspectives. The fusion module employs a devised attention mechanism to capture the semantic relevance between each two nodes' global and local embeddings. We summarize our contributions as follows:

---

[1] GLEN is available at `https://anonymous.4open.science/r/GLEN/`

- **New Finding.** We innovatively present that modeling from both global and local perspectives is indispensable for temporal graph representation learning. To the best of our knowledge, we are the first in the subfield of temporal graph learning to propose a method that simultaneously models the graph structure from an entire global perspective and a local subgraph perspective, and fuses all node embeddings across views.

- **New Method.** From the global perspective, we innovatively employ TCN instead of conventionally adopted RNNs for more stable and efficient training. From the local perspective, a new weighted sum algorithm based on time interval is devised to effectively aggregate neighborhood information. To better combine globally and locally acquired node embeddings, we introduce a cross-perspective fusion module based on a devised attention mechanism.

- **SOTA Performance.** Extensive experimental results on diverse real-world datasets for several predictive tasks demonstrate the advantages of GLEN. Moreover, multiple negative edge sampling strategies are employed for link prediction, which are proved to reflect real-world considerations for temporal graphs.

## 2 RELATED WORKS

**Static graph methods.** With a wide variety of applications, graph embedding has emerged as a focal point of increasing research interest (Zhou et al., 2020). Classical methods leverage matrix factorizations (Cao et al., 2015; Ou et al., 2016) or autoencoders (Pan et al., 2018; Hajiramezanali et al., 2019) to generate node embeddings. Random-walk-based methods such as DeepWalk (Perozzi et al., 2014), Node2Vec (Grover & Leskovec, 2016), LINE (Tang et al., 2015), and SDNE (Wang et al., 2016) employ a flexible and stochastic measure of node similarity and preserve the structural identity of nodes. Recent years have witnessed a burst of GNNs (graph neural networks) like GCN (Kipf & Welling, 2016), GAT (Velickovic et al., 2017), and GraphSAGE (Hamilton et al., 2017a) that automatically learn to encode graph structure by aggregating neighboring features.

**Temporal graph methods.** GNN-RNN-based temporal graph methods such as EvolveGCN (Pareja et al., 2020), CTGCN (Liu et al., 2020), and GCRN (Seo et al., 2018) learn constituent representations through GNNs in each snapshot and capture the temporal patterns across snapshots through RNNs. Message-passing temporal graph networks (MP-TGNs) such as JODIE (Kumar et al., 2019), TGAT (Xu et al., 2020), TGN (Rossi et al., 2020), APAN (Wang et al., 2021b), and TPGNN (Wang et al., 2022) aggregate local information through the message passing machnism. Walk-aggregating temporal graph networks (WA-TGNs) such as evolve2vec (Bastas et al., 2019a), STWalk (Pandhre et al., 2018), and EVONRL (Heidari & Papagelis, 2020) rely on temporal walks unfolding as the evolution of the graph. CAWN (Wang et al., 2021c) proposes causal anonymous walks using relative node identities. There are also some methods (Souza et al., 2022; Makarov et al., 2021) that leverage the advantages of both MP-TGNs and WA-TGNs.

## 3 PRELIMINARIES

A temporal graph contains a set of nodes: $\mathcal{V} = \{1, 2, \ldots, n\}$. To simplify the problem and be consistent with other works, we assume that the number of nodes in the graph remains constant (Wang et al., 2021b; Xu et al., 2020; Rossi et al., 2020). Throughout we use $l$ ($l \in \{0, 1, ..., L\}$) to denote the layer index of the network. Interaction events occur temporally between nodes, which is represented as an event stream $\mathcal{E} = \{\mathbf{e}_{uv}(t)\}$ ordered by time. $\mathbf{e}_{uv}(t)$ denotes a featured interaction between node $u$ and node $v$ at timestamp $t$ and is modeled as an edge in the graph. Each edge may disappear if it is not present in the dataset at some time. When two nodes interact at $t$, they are each other's temporal neighbor and multiple interactions can occur between each two nodes. We follow TGN (Rossi et al., 2020) to keep a memory module $\mathbf{s}_u(t) \in \mathbb{R}^d$ for each node $u$, which is a $d$-dimensional vector that summarizes the history information of $u$ and is updated as events occur. According to the message passing mechanism, when an interaction event $\mathbf{e}_{uv}(t)$ between $u$ and $v$ occurs at $t$, two messages are generated:

$$\mathbf{m}_u(t) = \text{msg}\left(\mathbf{s}_u(t^-), \mathbf{s}_v(t^-), \phi(t - t_u), \mathbf{e}_{uv}(t)\right),$$
$$\mathbf{m}_v(t) = \text{msg}\left(\mathbf{s}_v(t^-), \mathbf{s}_u(t^-), \phi(t - t_v), \mathbf{e}_{uv}(t)\right). \tag{1}$$

Here, $\mathbf{s}_u(t^-)$ denotes the memory of node $u$ just before $t$. $t_u$ is the time of the last event involving $u$. $\phi(\cdot)$ is a generic time encoding method (Xu et al., 2020; Rossi et al., 2020) that maps the time interval into a $d$-dimensional vector:

$$\phi(t) = [cos(\omega_1 t), sin(\omega_1 t), \ldots, cos(\omega_d t), sin(\omega_d t)], \tag{2}$$

where $\omega_i$ is learnable. msg is a message function such as concatenation or MLPs. Due to the batch processing in temporal graphs, all events involving node $u$ in a batch need to be aggregated as:

$$\overline{\mathbf{m}}_u(t) = \mathrm{agg}(\mathbf{m}_u(t_i) \mid t_i \leq t), \tag{3}$$

where agg is implemented by keeping only the most recent message for a given node $u$, which is the same as TGN-attn (Rossi et al., 2020). Then, the memory of node $u$ is updated as:

$$\mathbf{s}_u(t) = \mathrm{upd}\left(\overline{\mathbf{m}}_u(t), \mathbf{s}_u(t^-)\right), \tag{4}$$

where upd indicates a recurrent neural network (Chung et al., 2014). For another node $v$ involved in the event, its memory $\mathbf{s}_v(t)$ is updated in the same way.

## 4 PROPOSED METHOD

### 4.1 OVERALL FRAMEWORK

As shown in Figure 2, the framework of GLEN includes three major components: a GCN-TCN-based global embedding module, a local embedding module based on time interval weighting, and a cross-perspective fusion module. The global and local embedding modules respectively generate node embeddings from the global or local perspective. The cross-perspective fusion module is designed to effectively fuse the global and local node embeddings based on the attention mechanism, allowing the high-order information in a temporal graph (Liu et al., 2022) to be captured.

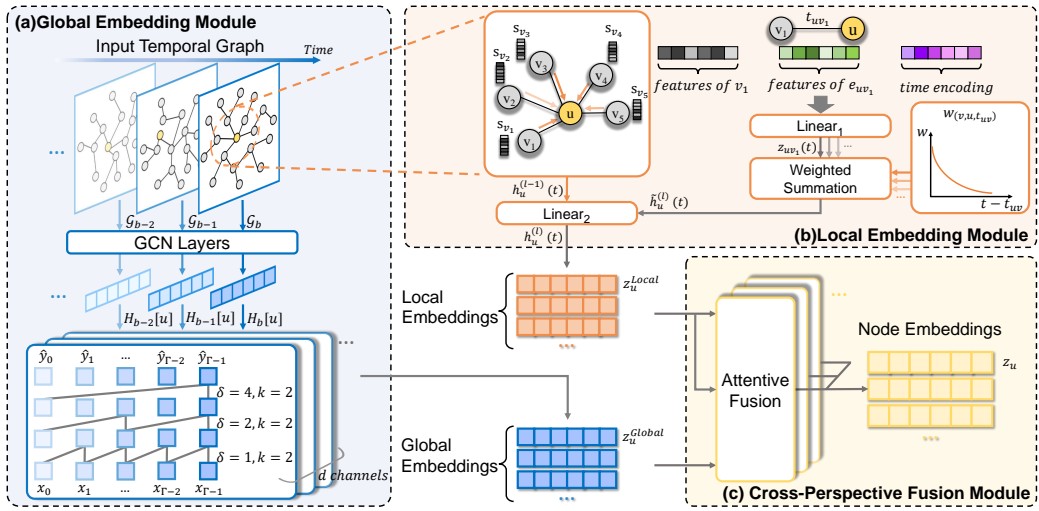

Figure 2: The overall framework of the proposed Global and Local Embedding Network (GLEN).

### 4.2 GCN-TCN-BASED GLOBAL EMBEDDING MODULE

The global embedding module of GLEN applies GCN (Kipf & Welling, 2016) to the graph composed of edges within a period (i.e., interaction events of a batch) to generate embeddings for graph nodes. The main reason for choosing GCN rather than other GNNs (e.g., GAT) is that GCN has higher computation efficiency. The obtained embedding matrices of several time steps are fed into TCN (Bai et al., 2018) to capture the temporal patterns of global graph evolution.

**Graph Convolutional Network (GCN).** Let $b$ denote the index of each batch and the events a batch occur at the same time step. The corresponding input of GCN includes the adjacency matrix $\mathbf{A}_b$ of

the graph consisting of edges in the $b$-th batch and the matrix of node features $\mathbf{X}_b \in \mathbb{R}^{n \times d}$. Since each row of $\mathbf{X}_b$ represents the attributes of a corresponding node, we take the sum of memory $\mathbf{s}_u$ and its temporal node features as the $d$-dimensional representation of node $u$ in $\mathbf{X}_b$. $\mathbf{s}_u$ indicates the updated node memory of $u$ after the events of the $b$-th batch, as temporal graph models assume that events of a single batch arrive simultaneously (Wang et al., 2021b).

GCN consists of $L$ layers of graph convolution. At each time step $b$, the $l$-th GCN layer takes $\mathbf{A}_b$ and the node embedding matrix $\mathbf{H}_b^l$ as input. The node embedding matrix is updated to $\mathbf{H}_b^{(l+1)}$ using the weight matrix $\mathbf{W}_b^{(l)}$. In each GCN layer, $\mathbf{A}_b$ is normalized to $\widehat{\mathbf{A}}_b$ first, defined as (for brevity, here we omit the subscript $b$):

$$\widetilde{\mathbf{A}} = \mathbf{A} + \mathbf{I}, \ \widetilde{\mathbf{D}} = \mathrm{diag}\left(\sum_v \widetilde{\mathbf{A}}_{uv}\right), \ \widehat{\mathbf{A}} = \widetilde{\mathbf{D}}^{-\frac{1}{2}}\widetilde{\mathbf{A}}\widetilde{\mathbf{D}}^{-\frac{1}{2}}, \tag{5}$$

where $\mathbf{I}$ is the identity matrix for adding self-loops and $\widetilde{\mathbf{D}}$ is the diagonal matrix used for propagating the features of each node's neighbors. Then the process of a single graph convolutional layer in GCN is described as the mathematical formula below:

$$\mathbf{H}_b^{(0)} = \mathbf{X}_b, \mathbf{H}_b^{(l+1)} = \sigma\left(\widehat{\mathbf{A}}_b\mathbf{H}_b^{(l)}\mathbf{W}_b^{(l)}\right), \tag{6}$$

where $\sigma(\cdot)$ is the relu activation function. The output of GCN is denoted as $\mathbf{H}_b^{(L)}$.

**Temporal Convolutional Network (TCN).** RNNs (Medsker & Jain, 1999; Chung et al., 2014; Hochreiter & Schmidhuber, 1997) generally suffer from inefficiency and unstable training (Ribeiro et al., 2020; Bengio et al., 1994). To avoid the problems, we innovatively adopt TCN (Bai et al., 2018) to model the sequential effect across snapshots, since it allows for parallel computation and uses techniques such as residual connection (He et al., 2016) and weight normalization (Salimans & Kingma, 2016) to make training more stable. For the output of GCN, we consider the chronological embeddings $\{\mathbf{H}_0^{(L)}[u], \mathbf{H}_1^{(L)}[u], ..., \mathbf{H}_b^{(L)}[u]\}$ of node $u$ as a temporal sequence with $d$ channels. To mitigate the so-called staleness problem, we set a time window constraint with a length of $\Gamma$ to limit the temporal range. Only the sequence elements of the $b$ and preceding $(\Gamma - 1)$ time steps are input into TCN. If the window size is 1, the current output of GCN is directly used as global node embeddings. For node $u$ and each channel $c \in \{1, 2, ..., d\}$, the input sequence of TCN is:

$$\mathcal{X} = \{x_0, x_1, ..., x_{\Gamma-1}\} = \{\mathbf{H}_{(b-\Gamma+1)}^{(L)}[u][c], \mathbf{H}_{(b-\Gamma+2)}^{(L)}[u][c], ..., \mathbf{H}_b^{(L)}[u][c]\}. \tag{7}$$

Then TCN applies the dilated convolution operation (Oord et al., 2016) on the sequence at each layer, and the formula is as follows:

$$\hat{y}_i = (\mathcal{X} *_\delta f)[i] = \sum_{j=0}^{k-1} f(j) \cdot x_{i-\delta \cdot j}, \tag{8}$$

where $*$ is the convolution operator, $k$ is the size of the filter $f : \{0, 1, ..., k-1\} \to \mathbb{R}$ and $\delta$ is the dilation factor of each layer increasing exponentially with the depth of TCN (i.e., at the $l$-th TCN layer, $\delta = 2^l$). TCN predicts the corresponding sequence $\{\hat{y}_0, \hat{y}_1, ..., \hat{y}_{\Gamma-1}\} = \mathrm{TCN}(\{x_0, x_1, ..., x_{\Gamma-1}\})$ and we take $\hat{y}_{\Gamma-1}$ as the output. The receptive field of one TCN layer is $(k-1) \times \delta$, thereby increasing the kernel size or using a deep network for a larger dilation factor enables richer historical information to be captured. Both the numbers of input channels and output channels of TCN are set to $d$. For node $u$, the outputs of $d$ kernels are considered the global embedding $\mathbf{z}_u^{\mathrm{Global}}$ that evolves over time. For time step $b$, the global embeddings of $n_b$ nodes involved in the events of the corresponding $b$-th batch are denoted as:

$$\mathbf{Z}^{\mathrm{Global}} \in \mathbb{R}^{n_b \times d} = \{\mathbf{z}_1^{\mathrm{Global}}, \mathbf{z}_2^{\mathrm{Global}}, ..., \mathbf{z}_{n_b}^{\mathrm{Global}}\}. \tag{9}$$

### 4.3 Local Embedding Module Based on Time Interval Weighting

This module generates the local embedding $\mathbf{z}_u^{\mathrm{Local}}$ that evolves over time for each node $u$ from the local perspective. There is a common pattern in temporal graphs: recent events tend to have more important potential information. Therefore, we devise a weighted sum algorithm based on time interval to effectively aggregate the information of temporal neighbors.

To control the computation cost and ensure a fair comparison, we restrict the neighborhood size of each node like other works (Rossi et al., 2020; Wang et al., 2021c). We denote the neighbor set of $u$ at $t$ as $\mathcal{N}_u(t)$, which contains a certain number $|\mathcal{N}|$ of the most recent neighbors that interact with $u$ before $t$. If the timestamp $t_{uv}$ of the event $\mathbf{e}_{uv}$ is far from the current time $t$, the impact of $\mathbf{e}_{uv}$ and $v$ on node $u$ should be reduced. Thus, different temporal weights for neighbors are computed as:

$$w_{(v,u,t)} = \frac{\exp(-(t - t_{uv}))}{\sum_{(v',t_{uv'})\in\mathcal{N}_u(t)} \exp(-(t - t_{uv'}))}. \tag{10}$$

The temporal weight decreases as the time interval increases. We generate the corresponding representation vector for the neighbor $v$ and event $\mathbf{e}_{uv}$ through a linear layer:

$$\mathbf{z}_{uv}^{(l)}(t) = \text{Tanh}\left(\text{Linear}_1(\mathbf{h}_v^{(l-1)}(t)\|\mathbf{e}_{uv}(t)\|\phi(t - t_{uv}))\right), \tag{11}$$

where $\mathbf{h}_v^{(l-1)}(t)$ is the input of the $l$-th network layer, and $\mathbf{h}_v^{(0)}(t)$ is the sum of $\mathbf{s}_v(t)$ and temporal node features. The activation function $\text{Tanh}$ is used to provide nonlinear transformations and limit the values in a certain range to facilitate the subsequent summation. We then utilize the temporal weights to aggregate neighborhood information for node $u$ through a weighted sum:

$$\tilde{\mathbf{h}}_u^{(l)}(t) = \sum_{(v,t_{uv})\in\mathcal{N}_u(t)} w_{(v,u,t)} \cdot \mathbf{z}_{uv}^{(l)}(t). \tag{12}$$

The node embedding of $u$ is generated by combining its own representation with aggregated neighborhood information through a linear layer:

$$\mathbf{h}_u^{(l)}(t) = \text{Linear}_2(\mathbf{h}_u^{(l-1)}(t)\|\tilde{\mathbf{h}}_u^{(l)}(t)). \tag{13}$$

After all events of the $b$-th batch are processed, the output of the module is taken as the local embedding of node $u$: $\mathbf{z}_u^{\text{Local}} = \mathbf{h}_u^{(L)}(t)$ that evolves over time. Similar to Eq. 9, the local embedding matrix of the $b$-th time step is denoted as:

$$\mathbf{Z}^{\text{Local}} \in \mathbb{R}^{n_b \times d} = \left\{\mathbf{z}_1^{\text{Local}}, \mathbf{z}_2^{\text{Local}}, \dots, \mathbf{z}_{n_b}^{\text{Local}}\right\}. \tag{14}$$

## 4.4 CROSS-PERSPECTIVE FUSION MODULE

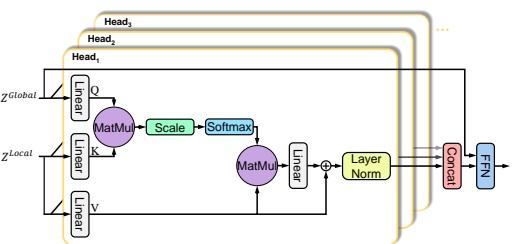

Figure 3: The cross-perspective fusion module of GLEN calculates embeddings $\mathbf{Z}$ according to the relevance between $\mathbf{z}_u^{\text{Global}}$ and $\mathbf{z}_v^{\text{Local}}$ of each two nodes $u$ and $v$.

The number of GNN layers is generally limited, and neighborhood-based aggregation impairs the ability to propagate messages to distant nodes, which means only short-range signals can be captured (Mao et al., 2023). Although the longer walk length of WA-TGNs indicates the deeper depth of the sampled neighborhood, some studies indicate that longer random walks do not necessarily imply better performance (Talati et al., 2021). WA-TGNs such as CAWN (Wang et al., 2021c) only use short walk lengths, preventing long-range dependencies in graphs from being captured. To extract high-order features in the graph and further yield semantic proximities between node embeddings, we devise a cross-perspective fusion module based on attention (Vaswani et al., 2017a) for GLEN to combine global and local node embeddings, as illustrated in Figure 3. Multi-head attention allows the model to jointly attend to crucial information in different representation subspaces. We use $\eta$ to indicate the number of heads and $i$ to denote the index of each head. In each single attention head, we forward global embeddings to a linear projection to obtain the 'query', and local embeddings to another two different linear projections to obtain the 'key' and 'value':

$$\mathbf{Q}_i = \mathbf{Z}^{\text{Global}}\mathbf{W}_i^Q, \ \mathbf{K}_i = \mathbf{Z}^{\text{Local}}\mathbf{W}_i^K, \ \mathbf{V}_i = \mathbf{Z}^{\text{Local}}\mathbf{W}_i^V, \tag{15}$$

where $\mathbf{W}_i^Q \in \mathbb{R}^{d \times d_k}, \mathbf{W}_i^K \in \mathbb{R}^{d \times d_k}, \mathbf{W}_i^V \in \mathbb{R}^{d \times d_v}$ are three transformation matrices and $d_k = d_v = d/\eta$. The attentive output of each head is denoted as:

$$\tilde{\mathbf{Z}}_i = \text{softmax}(\frac{\mathbf{Q}_i\mathbf{K}_i^T}{\sqrt{d_k}})\mathbf{V}_i. \tag{16}$$

The attention coefficient of each two nodes $u$ and $v$ implies the correlation between $\mathbf{z}_u^{\text{Global}}$ and $\mathbf{z}_v^{\text{Local}}$ and increases with relevance. The outputs of all heads are concatenated as the output of the attention mechanism:

$$\tilde{\mathbf{Z}} = \text{MultiHead}(\mathbf{Q}, \mathbf{K}, \mathbf{V}) = \text{Concat}(\tilde{\mathbf{Z}}_1, \tilde{\mathbf{Z}}_2, ..., \tilde{\mathbf{Z}}_\eta)\mathbf{W}^O, \tag{17}$$

where $\mathbf{W}^O \in \mathbb{R}^{\eta d_v \times d}$. Since GLEN chooses the linear projection of local embeddings as the 'value', $\tilde{\mathbf{Z}}$ actually gives hidden representations of the local embeddings. To further combine these latent representations with global node embeddings, we concatenate $\tilde{\mathbf{z}}_u \in \tilde{\mathbf{Z}}$ with $\mathbf{z}_u^{\text{Global}}$ and input them to a feedforward neural network to further capture the nonlinear correlation between local and global embeddings of the same node:

$$\mathbf{z}_u = \text{FFN}(\tilde{\mathbf{z}}_u || \mathbf{z}_u^{\text{Global}}). \tag{18}$$

The attention mechanism captures the correlation between each two nodes, allowing for the retention of high-order information in temporal graphs. The intent representations influenced by the affinity weight matrix enable the model to selectively focus on pairs of nodes with high relevance and ignore mostly unimportant information.

To empirically prove that both global and local perspectives are important and enhance the interpretability of our fusion module, we additionally conduct a case study on the correlation of node embeddings in Appendix A. Experimental results reveal that both global and local views are essential since considering only one is not comprehensive.

## 5 EXPERIMENTS

### 5.1 DATASETS AND BASELINES

We totally use seven public real-world temporal graph datasets to extensively validate the effectiveness of GLEN, including Wikipedia (Kumar et al., 2019), Reddit (Kumar et al., 2019), Enron (Shetty & Adibi, 2004), UCI (Panzarasa et al., 2009), UN Trade (MacDonald et al., 2015), MOOC (Kumar et al., 2019), and Flights (Schäfer et al., 2014). Descriptions and statistics of the datasets are reported in Appendix C.1. We choose eight state-of-the-art approaches for temporal graph representation learning as strong baselines to compare with, including DyRep (Trivedi et al., 2019), JODIE (Kumar et al., 2019), TGAT (Xu et al., 2020), TGN (Rossi et al., 2020), CAWN (Wang et al., 2021c), PINT (Souza et al., 2022), GraphMixer (Cong et al., 2023), and TIGER (Zhang et al., 2023). Introductions of baselines are available in Appendix C.2. For the settings of baselines, we use their recommended configurations. We use the same data processing and splitting procedures as TGAT (Xu et al., 2020) and TGN (Rossi et al., 2020). For fairness, we evaluate all the methods in the same environment and on the same Nvidia Tesla V100-SXM2 GPU to obtain experimental results.

### 5.2 IMPLEMENTATION DETAILS AND EVALUATION PROTOCOL

We conduct experiments on two predictive tasks: link prediction (Zhang et al., 2020; Srinivasan & Ribeiro, 2019; Lü & Zhou, 2011) and dynamic node classification (Aggarwal & Li, 2011; Xu et al., 2019). For all datasets, we split edges chronologically by 70%, 15%, and 15% for training, validation, and testing. We use the Adam optimizer and early stopping with a patience of 5 for training. For both link prediction and dynamic node classification, we use BCE loss. All the settings are consistent with those set by baselines (Xu et al., 2020; Rossi et al., 2020; Wang et al., 2021c). More implementation details of GLEN can be found in Appendix C.3. The pseudo-code of GLEN can be seen in Appendix C.4. In addition, random, historical, and inductive negative sampling strategies (Poursafaei et al., 2022) are applied for evaluation, which proves to reflect real-world considerations for temporal graphs. More details of the evaluation protocol are introduced in Appendix C.5.

### 5.3 QUANTITATIVE RESULTS

Table 2 presents the results of inductive link prediction experiments, where NS is the abbreviation of negative sampling. Since PINT takes too long on the largest Flights dataset, we did not include PINT's results on Flights in Table 2. In the link prediction task, GLEN obviously outperforms the baselines on all datasets under the evaluation of all negative sampling strategies.

Baselines have a significant performance drop with both historical and inductive sampling strategies, as their one-sidedness makes it difficult to correctly predict the pattern of interactive appearance in temporal graphs. Whereas, GLEN can keep the performance relatively stable owing to the complementarity between information acquired globally and locally, and the effective cross-perspective fusion. The results of dynamic node classification are shown in Table 1, where GLEN also obtains the best results on all datasets. The results of transductive link prediction are reported in Appendix D.1, where GLEN also shows the state-of-the-art performance.

Table 1: Average ROC AUC (%) of dynamic node classification (over 5 runs). (**First** second)

| Methods | Wikipedia | Reddit | MOOC |
|---|---|---|---|
| DyRep | 80.79±1.86 | 50.01±2.27 | 66.08±0.24 |
| JODIE | 84.46±2.84 | 61.57±4.34 | 69.46±0.51 |
| TGAT | 85.98±1.45 | 65.87±1.45 | 54.05±0.20 |
| TGN | 87.33±0.30 | 60.09±1.64 | 64.09±0.68 |
| GraphMixer | 86.26±1.83 | 63.24±1.91 | 68.65±1.09 |
| TIGER | 85.55±0.30 | 68.83±1.62 | 70.99±0.05 |
| **GLEN** | **90.16±0.32** | **70.21±0.27** | **71.49±0.33** |

Table 2: Average Precision (%) of link prediction under different negative sampling strategies in the inductive setting (over 5 runs). (**First** second)

| NS Strategy | Methods | Wikipedia | Reddit | Enron | UCI | UN Trade | MOOC | Flights |
|---|---|---|---|---|---|---|---|---|
| Random | DyRep | 72.17±1.14 | 55.13±1.04 | 59.99±3.77 | 60.55±1.29 | 59.22±0.75 | 64.21±0.59 | 92.47±0.72 |
| | JODIE | 97.97±0.00 | 99.26±0.74 | 81.68±0.10 | 98.06±0.23 | 57.96±6.18 | 83.23±6.50 | 94.85±0.64 |
| | TGAT | 94.03±0.20 | 96.62±0.15 | 56.01±2.46 | 74.39±5.33 | 59.80±0.83 | 71.21±0.41 | 89.02±0.06 |
| | TGN | 98.00±0.18 | 94.09±1.07 | 75.28±3.37 | 83.04±2.37 | 57.42±1.73 | 81.51±3.31 | 84.11±0.61 |
| | CAWN | 99.64±0.34 | 99.82±0.10 | 92.05±1.77 | 98.45±0.66 | 91.64±0.92 | 87.59±1.88 | 98.67±0.14 |
| | PINT | 98.50±0.08 | 97.90±0.39 | 88.29±0.23 | 96.20±0.25 | 69.10±0.60 | 96.11±4.15 | - |
| | GraphMixer | 96.49±0.08 | 95.22±0.03 | 75.67±0.55 | 90.79±0.32 | 56.47±2.82 | 80.95±0.65 | 83.00±0.07 |
| | TIGER | 98.30±0.02 | 98.64±0.53 | 83.40±1.13 | 92.98±0.23 | 55.29±0.11 | 84.72±1.48 | 91.84±0.86 |
| | **GLEN** | **99.95±0.05** | **99.85±0.28** | **96.15±1.61** | **99.11±0.30** | **96.09±0.12** | **96.48±4.02** | **99.36±0.17** |
| Historical | DyRep | 69.45±1.16 | 52.40±1.68 | 56.96±3.12 | 52.67±0.87 | 59.55±0.81 | 60.93±0.58 | 62.00±1.81 |
| | JODIE | 40.64±0.32 | 49.68±0.22 | 51.26±0.67 | 54.23±2.32 | 58.07±2.47 | 47.14±5.81 | 60.41±2.39 |
| | TGAT | 71.35±0.93 | 63.25±0.78 | 53.45±2.52 | 61.62±0.49 | 51.85±3.03 | 59.60±0.59 | 64.43±0.32 |
| | TGN | 81.96±1.10 | 61.99±1.30 | 61.90±2.01 | 72.31±1.54 | 54.41±1.00 | 63.70±2.02 | 58.27±1.73 |
| | CAWN | 80.14±8.52 | 82.10±1.33 | 58.58±4.36 | 81.13±12.5 | 87.10±0.75 | 97.23±4.22 | 51.84±0.13 |
| | PINT | 64.97±7.12 | 68.27±7.53 | 78.66±0.68 | 84.73±0.91 | 58.50±6.18 | 67.35±4.25 | - |
| | GraphMixer | 88.02±0.39 | 64.48±0.36 | 73.18±1.20 | 80.29±0.31 | 58.92±2.67 | 74.07±0.73 | 65.23±0.23 |
| | **GLEN** | **96.25±0.27** | **97.31±2.46** | **97.28±0.53** | **95.94±2.00** | **95.78±2.32** | **99.53±0.93** | **76.96±0.54** |
| Inductive | DyRep | 69.36±1.20 | 52.48±1.41 | 57.16±3.34 | 52.68±0.90 | 59.57±0.90 | 60.92±0.62 | 61.99±1.81 |
| | JODIE | 40.58±0.18 | 49.73±0.16 | 51.46±0.42 | 54.16±2.58 | 57.88±2.56 | 47.15±5.77 | 59.47±1.36 |
| | TGAT | 71.46±0.79 | 63.29±0.64 | 53.98±3.02 | 62.66±0.84 | 51.94±2.83 | 59.65±0.68 | 64.42±0.32 |
| | TGN | 81.90±1.28 | 62.15±1.46 | 62.37±2.47 | 72.25±1.55 | 54.48±1.07 | 63.61±2.00 | 58.14±1.77 |
| | CAWN | 68.70±1.48 | 78.34±1.37 | 62.22±6.60 | 83.32±7.21 | 89.83±1.64 | 90.93±1.38 | 53.48±0.04 |
| | PINT | 64.86±7.09 | 72.79±5.75 | 78.59±0.73 | 84.72±1.03 | 54.39±1.88 | 67.36±4.33 | - |
| | GraphMixer | 83.91±0.54 | 63.96±0.26 | 73.19±1.19 | 80.33±0.31 | 58.89±2.66 | 74.08±0.73 | 63.13±0.15 |
| | **GLEN** | **96.13±0.29** | **97.28±2.48** | **97.38±0.46** | **95.43±2.63** | **95.76±2.32** | **99.54±0.91** | **77.23±0.61** |

## 5.4 EFFICIENCY

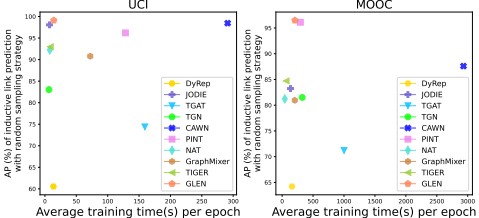

Figure 4: Trade-off between AP (%) in link prediction and training time.

We further evaluate the ability to trade off the precision and efficiency of GLEN, which is illustrated in Figure 4. The AP (Average Precision) is computed with the random negative sampling strategy and the inductive setting in a percentage format. Methods closer to the upper left corner of the figure are more ideal. Note that the training time of PINT here does not include precomputing the positional features, otherwise its training time will be unbearably long. The efficiency of GLEN is comparable to the fastest baselines, and the performance is improved. The complexity analysis and more experimental results about efficiency are reported in Appendix B and Appendix D.2 respectively. Overall, GLEN strikes an impressive balance between inference precision and training speed, which can be attributed to the training efficiency of TCN.

## 5.5 HYPER-PARAMETER INVESTIGATION

We systematically analyze the effect of hyper-parameters related to GLEN, including the time window size $\Gamma$, number of sampled neighbors $|\mathcal{N}|$, number of layers in TCN, kernel size of TCN, number of heads $\eta$ in the multi-head attention mechanism, and the dropout ratio. Figure 5 illustrates the impact of various hyper-parameters on GLEN. We combine the hyper-parameters in pairs.

The reason for jointly considering dropout and attention heads is that they both mainly affect the cross-perspective fusion module of GLEN. Both the number of layers in TCN and the kernel size of TCN affect the receptive field and the temporal convolution operations of TCN, so we consider them together. GLEN exhibits its robustness as the fluctuation of AP is small. The effect of $\Gamma$ and $|\mathcal{N}|$ on GLEN is shown in Figure 6. An interesting insight is that GLEN tends to achieve the maximum AP with a small time window size, which means crucial recent infor-

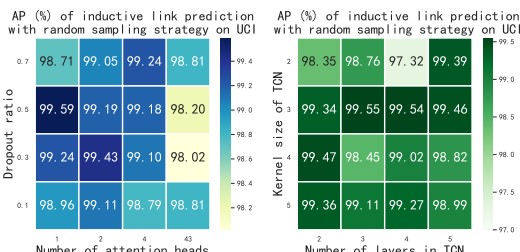

Figure 5: Performance of GLEN on the UCI dataset with different hyper-parameters.

mation is sufficient for GLEN to capture the evolution patterns of tamporal graphs. $|\mathcal{N}|$ barely makes a difference to GLEN, while other local-view TGNs methods typically require a certain number (usually 10 or 20) of neighbor nodes to achieve their best performance (Rossi et al., 2020; Wang et al., 2021c). This indicates that global embeddings supplement local embeddings through GLEN's fusion module. More experimental results on hyper-parameter investigation are reported in Appendix D.3.

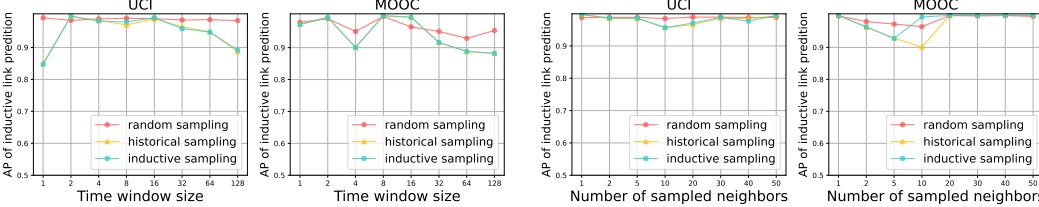

Figure 6: Performance of GLEN with different time window sizes and numbers of sampled neighbors with different negative sampling strategies.

## 5.6 ABLATION STUDY

We further analyze GLEN by performing an ablation study to manifest the contributions of different components of GLEN. More details of the ablation study are reported in Appendix C.6. We summarize the results of the ablation study on link prediction in Table 3. From the results, we can observe that removing any of GLEN's components will cause performance degradation, indicating that the modules we designed are indispensable for temporal graph representation learning. The ablation study further proves the effectiveness of the cross-perspective fusion module and provides a certain degree of interpretability for the complementarity between global and local modeling of temporal graphs. Results of the ablation study on dynamic node classification are reported in Appendix D.4.

Table 3: Average Precision (%) for ablation study of GLEN in inductive link prediction.

| Ablation | Enron | | | UCI | | | UN Trade | | | MOOC | | |
|---|---|---|---|---|---|---|---|---|---|---|---|---|
| | Random | Historical | Inductive | Random | Historical | Inductive | Random | Historical | Inductive | Random | Historical | Inductive |
| w/o GCN | 85.87 | 73.74 | 75.09 | 96.28 | 95.42 | 95.42 | 91.61 | 91.72 | 91.73 | 87.26 | 84.88 | 84.88 |
| w/o TCN | 87.40 | 70.09 | 71.23 | 95.36 | 81.97 | 81.97 | 95.79 | 95.33 | 95.32 | 94.87 | 97.02 | 97.01 |
| w/o Global | 84.62 | 73.21 | 75.28 | 81.20 | 70.01 | 70.01 | 84.00 | 84.35 | 84.37 | 75.15 | 67.09 | 67.09 |
| w/o Local | 90.58 | 93.00 | 93.07 | 80.31 | 70.95 | 70.95 | 90.45 | 90.59 | 90.70 | 62.66 | 56.29 | 56.29 |
| w/o Fusion | 85.55 | 89.75 | 89.78 | 81.16 | 69.23 | 69.23 | 90.95 | 91.07 | 91.07 | 77.42 | 66.43 | 66.43 |
| GLEN | **96.15** | **97.28** | **97.38** | **98.47** | **95.94** | **95.43** | **96.09** | **95.78** | **95.76** | **96.48** | **99.53** | **99.54** |

## 6 CONCLUSION

In this paper, we proposed the Global and Local Embedding Network (GLEN), an adventurous method for temporal graph representation learning. Specifically, GLEN consists of three main components: the GCN-TCN-based global embedding module, the local embedding module based on time interval weighting, and the cross-perspective fusion module. The global embedding module models temporal graphs from a global perspective, while the local embedding module does so from a local perspective. Then, the fusion machanism combines global and local embeddings based on a novel attention mechanism. By taking both global and local perspectives, GLEN outperforms all the baselines in extensive experiments.

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

## A INTERPRETABILITY

In order to empirically prove that both global and local perspectives are important as motivation and enhance the interpretability of our method, we additionally conduct a case study on the correlation of node embeddings by multiplying them through the inner-product to obtain the correlation matrix. Link prediction is the main task of temporal graph representation learning. The way to improve the performance of link prediction is to learn high-quality embeddings for graph nodes such that the correlation between the embeddings of the node pairs that will establish a connection is higher. In Figure 7 and Figure 8, we show this in the form of heat maps. The deeper color indicates a stronger correlation and a higher connection probability. In Figure 7, associated events are (1,2), (3,4), and (5,2). With the exception of each node itself, the global embeddings of nodes $2, 5$ have the maximum correlation. However, the global embeddings cannot reflect the associations between nodes $1, 2$ and nodes $3, 4$. Similarly, the local embeddings of nodes $1, 2$ and nodes $3, 4$ have strong correlations, but local embeddings cannot reveal the association between nodes $2, 5$. After the process of the Cross-Perspective Fusion Module of GLEN, the strong correlations between all node pairs can be reflected through fused embeddings. In Figure 8, associated events are (9,10), (9,11), and (12,13). This case can be analyzed similarly to Figure 7. Global and local embeddings can capture some correlations between linked nodes, but also ignore some. The strong correlations between all linked nodes can be reflected in fused embeddings. Thus both global and local views are essential as motivation since considering only one is not comprehensive.

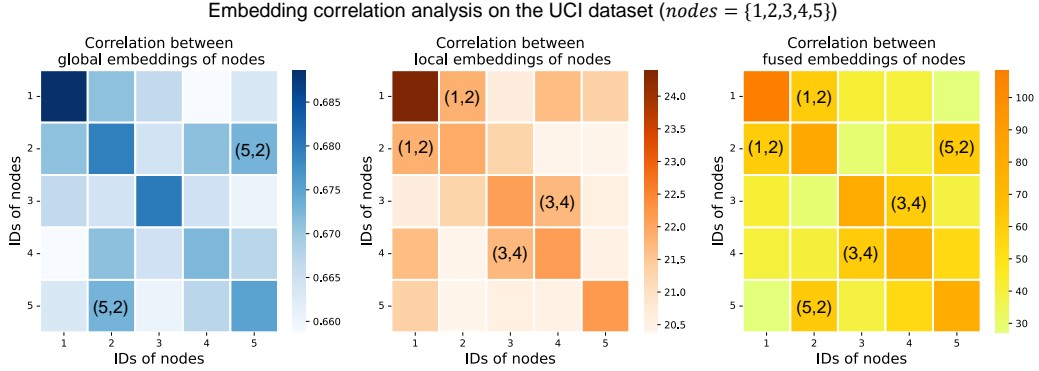

Figure 7: Case study about the correlation of node embeddings on UCI. Associated nodes are $\{1, 2, 3, 4, 5\}$. Associated events are (1,2), (3,4), and (5,2). The deeper color indicates a stronger correlation and a higher connection probability.

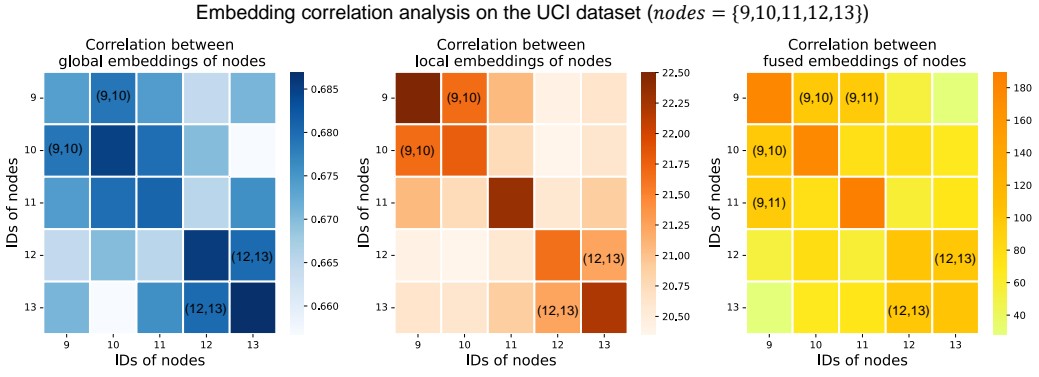

Figure 8: Case study about the correlation of node embeddings on UCI. Associated nodes are $\{9, 10, 11, 12, 13\}$. Associated events are (9,10), (9,11), and (12,13). The deeper color indicates a stronger correlation and a higher connection probability.

## B  Time Complexity Analysis

In order to adequately understand the complexity of different temporal graph learning methods and thus analyze why GLEN achieves impressive efficiency compared to recent works despite employing multiple modules, we perform a detailed complexity analysis of training different models. We mainly focus on the computational complexity of the core procedures of each method, omitting the time spent on inconsequential operations such as time encoding, linear transformations, MLPs, etc. Throughout we use $T$ to denote the number of time steps (batches) and $|\mathcal{E}|$ to denote the number of events that have been processed. The number of nodes in the graph is denoted as $n = |\mathcal{V}|$.

### B.1  Time Complexity of MP-TGNs

To fix the computational pattern, MP-TGNs generally sample $|\mathcal{N}|$ neighbors for each node when an event involving it occurs, and we use $\mathcal{C}_{\text{Sampling}}$ to denote the time complexity required for each sampling operation. Common sampling strategies include the most recent neighbor sampling (Rossi et al., 2020), uniform sampling (Rossi et al., 2020), time decay sampling (Zheng et al., 2021), and gumbel sampling (Zheng et al., 2021).

Next, MP-TGNs aggregate information for each node $u$ from its $|\mathcal{N}|$ neighbors. The information disseminated includes features of neighboring nodes, features of edges, and time encoding vectors. We denote the time complexity of message passing as $\mathcal{C}_{\text{Messge Passing}}$. For example, TGN (Rossi et al., 2020) uses the multi-head attention (Vaswani et al., 2017b) mechanism for information propagation. The 'query' is the linear projection of the concatenation of the node's own feature vector as well as the time encoding vector. The 'key' and 'value' are both the linear projection of concatenations of $|\mathcal{N}|$ neighboring nodes's features, the features of corresponding edges, and time encodings. We use $\eta$ to denote the number of heads and $d$ to denote the dimension of 'queue', 'key', and 'value'. Then, in each head, $\mathbf{Q}\mathbf{K}^T$ is first calculated, where $\mathbf{Q} \in \mathbb{R}^{1 \times (d/\eta)}$ and $\mathbf{K} \in \mathbb{R}^{|\mathcal{N}| \times (d/\eta)}$. The corresponding computational complexity is:

$$O(|\mathcal{N}| \cdot d/\eta). \tag{19}$$

The time complexity of the following scaling operation and the softmax function is $O(|\mathcal{N}|)$. Then the obtained weight matrix is multiplied by $\mathbf{V} \in \mathbb{R}^{|\mathcal{N}| \times (d/\eta)}$, and the time complexity is also $O(|\mathcal{N}| \cdot d/\eta)$. Therefore, the total time complexity of the multi-head attention mechanism is:

$$
\begin{aligned}
&(O(|\mathcal{N}| \cdot d/\eta) + O(|\mathcal{N}|) + O(|\mathcal{N}| \cdot d/\eta)) \cdot \eta \\
&= O(|\mathcal{N}| \cdot d/\eta) \cdot \eta \\
&= O(|\mathcal{N}| \cdot d).
\end{aligned} \tag{20}
$$

That is, the estimated time complexity of the message passing operation of TGN (Rossi et al., 2020) is:

$$\mathcal{C}_{\text{Message Passing}}^{\text{TGN}} = O(|\mathcal{N}| \cdot d). \tag{21}$$

Since the sampling and message passing operations are performed for each element in the event stream, the time complexity of MP-TGNs can be calculated as:

$$\mathcal{C}_{\text{MP-TGNs}} = |\mathcal{E}| \cdot (\mathcal{C}_{\text{Sampling}} + \mathcal{C}_{\text{Message Passing}}). \tag{22}$$

Specifically, we denote the time spent on each neighbor sampling operation of TGN as $\mathcal{C}_{\text{Sampling}}^{\text{TGN}}$. Then the time complexity of TGN is estimated as:

$$
\begin{aligned}
\mathcal{C}_{\text{TGN}} &= |\mathcal{E}| \cdot (\mathcal{C}_{\text{Sampling}}^{\text{TGN}} + \mathcal{C}_{\text{Message Passing}}^{\text{TGN}}) \\
&= |\mathcal{E}| \cdot (\mathcal{C}_{\text{Sampling}}^{\text{TGN}} + O(|\mathcal{N}| \cdot d)).
\end{aligned} \tag{23}
$$

### B.2  Time Complexity of WA-TGNs

WA-TGNs also sample the neighbors of each node involved in an event. We also use $\mathcal{C}_{\text{Sampling}}$ to denote the time complexity required for each sampling operation. WA-TGNs encode the sampled temporal walks to generate representations of nodes, and the corresponding model complexity depends mainly on the walk length and the number of walks. In fact, WA-TGNs need to perform

multiple wanderings for each node over the time span and select an exact neighbor node at each wandering step. Therefore, the time complexity of WA-TGNs is extraordinarily high. For the sampled walks, different methods encode walks in different ways. Here we take CAWN (Wang et al., 2021c) as a specific example and denote the time spent on each neighbor sampling operation of CAWN as $\mathcal{C}_{\text{Sampling}}^{\text{CAWN}}$. It is worth noting that CAWN samples a set of $M$ walks for each node to anonymize the walks by generating relative node identities. We denote the length of each walk as $m$, the same as the official paper of CAWN. Then the complexity of sampling a walk set is $O(M \cdot m \cdot \mathcal{C}_{\text{Sampling}}^{\text{CAWN}})$, which means sampling each step of the $M$ walks. Then, CAWN anonymizes a target walk with $O(M \cdot m)$ complexity since the relative node identities are obtained based on the counts that nodes appear at a certain position according to the sampled walks of the set. CAWN uses a simple RNN to encode each anonymized walk with $O(m)$ time complexity. Then the time complexity to complete the sampling, anonymization, and encoding of $M$ walks in a set is:

$$
\begin{aligned}
& O(M \cdot m \cdot \mathcal{C}_{\text{Sampling}}^{\text{CAWN}}) + M \cdot (O(M \cdot m) + O(m)) \\
=& O(M \cdot m \cdot (\mathcal{C}_{\text{Sampling}}^{\text{CAWN}} + M)).
\end{aligned}
\tag{24}
$$

For each event, the two encoded walk sets of the involved nodes are aggregated with time complexity $O(M \cdot m)$. Similar to MP-TGNs, the sampling and walk encoding operations are performed for each element in the event stream. Thus, the time complexity of CAWN can be computed as:

$$
\begin{aligned}
\mathcal{C}_{\text{CAWN}} &= |\mathcal{E}| \cdot (O(M \cdot m \cdot (\mathcal{C}_{\text{Sampling}}^{\text{CAWN}} + M)) + O(M \cdot m)) \\
&= |\mathcal{E}| \cdot O(M \cdot m \cdot (\mathcal{C}_{\text{Sampling}}^{\text{CAWN}} + M)).
\end{aligned}
\tag{25}
$$

### B.3 TIME COMPLEXITY OF GLEN

#### B.3.1 TIME COMPLEXITY OF THE GLOBAL EMBEDDING MODULE

According to the formulas of GCN (Kipf & Welling, 2016) in Section 3.2, the time complexity of GCN is related to the number of nodes $n$, the dimension of node features $d$, and the number of GCN layers $L_{\text{GCN}}$. The time complexity formula of GCN is (Chiang et al., 2019):

$$
\mathcal{C}_{\text{GCN}} = O(n \cdot d^2 \cdot L_{\text{GCN}}).
\tag{26}
$$

Here we assume that the graph is sparse, thus ignoring the computational overhead of normalizing the adjacency matrix. The time complexity of TCN (Bai et al., 2018) is related to the number of TCN layers $L_{\text{TCN}}$, the size of convolutional kernels $k$, the number of channels of each convolutional kernel (i.e., the number of input channels $d$), and the number of convolutional kernels (i.e., the number of output channels $d$). According to Section Local Embedding Module Based on Time Interval Weighting, in each layer of TCN, the number of convolution operations for each 1-D convolution kernel is $\frac{\Gamma - 1}{(k-1) \cdot \delta}$. The time complexity of each kernel's convolution operation is $O(k \cdot d)$ and there are $d$ kernels. The expansion factor of the $l$-th TCN layer is $\delta = 2^l$, so the time complexity of TCN is:

$$
\begin{aligned}
\mathcal{C}_{\text{TCN}} &= \sum_{l=0}^{L_{\text{TCN}}-1} O(k \cdot d) \cdot \frac{\Gamma - 1}{(k-1) \cdot \delta} \cdot d \\
&= \sum_{l=0}^{L_{\text{TCN}}-1} O(\frac{\Gamma}{2^l} \cdot d^2) \\
&= O(\Gamma \cdot (1 - 2^{-L_{\text{TCN}}}) \cdot d^2).
\end{aligned}
\tag{27}
$$

If we use an RNN (Medsker & Jain, 1999) with $L_{\text{RNN}}$ layers to update the feature matrix of nodes at each time step, the time complexity can be estimated as:

$$
\mathcal{C}_{\text{RNN}} = O(T \cdot n^2 \cdot d \cdot L_{\text{RNN}}).
\tag{28}
$$

By comparing Eq.27 and Eq.28, it can be seen that the time complexity of TCN is reduced compared to RNN, and TCN also eliminates the time-dependent problem of RNN. So TCN can be easily parallelized and computed more efficiently using GPUs. The time complexity of GLEN's global embedding module can be computed as:

$$
\begin{aligned}
\mathcal{C}_{\text{Global}} &= T \cdot (\mathcal{C}_{\text{GCN}} + \mathcal{C}_{\text{TCN}}) \\
&= T \cdot (O(n \cdot d^2 \cdot L_{\text{GCN}}) + O(\Gamma \cdot (1 - 2^{-L_{\text{TCN}}}) \cdot d^2)) \\
&= T \cdot O(d^2 \cdot (n \cdot L_{\text{GCN}} + \Gamma \cdot (1 - 2^{-L_{\text{TCN}}}))).
\end{aligned}
\tag{29}
$$

### B.3.2 Time Complexity of the Local Embedding Module

In the local embedding module of GLEN, we use the time-saving weighted sum for neighborhood aggregation, which only requires a time complexity of $O(|\mathcal{N}|)$. Thus, the complexity of GLEN's local embedding module is:

$$\mathcal{C}_{\text{Local}} = |\mathcal{E}| \cdot (\mathcal{C}_{\text{Sampling}}^{\text{GLEN}} + O(|\mathcal{N}|)), \tag{30}$$

where $\mathcal{C}_{\text{Sampling}}^{\text{GLEN}}$ is the complexity of each neighbor sampling operation of GLEN. The sampling strategy of GLEN is the most recent neighbor sampling, which is not very time consuming.

### B.3.3 Time Complexity of the Fusion Module

Similar to the time complexity of the multi-head attention mechanism computed in Appendix B.1, the time complexity of GLEN's fusion module is:

$$\mathcal{C}_{\text{Fusion}} = T \cdot O(n_b^2 \cdot d), \tag{31}$$

where $n_b$ is the number of nodes involved in the events of a batch. By summing up Eq.29, Eq.30, and Eq.31, the time complexity of GLEN can be computed as:

$$\begin{aligned}
\mathcal{C}_{\text{GLEN}} =& \mathcal{C}_{\text{Global}} + \mathcal{C}_{\text{Local}} + \mathcal{C}_{\text{Fusion}} \\
=& T \cdot O(d^2 \cdot (n \cdot L_{\text{GCN}} + \Gamma \cdot (1 - 2^{-L_{\text{TCN}}}))) \\
& + |\mathcal{E}| \cdot (\mathcal{C}_{\text{Sampling}}^{\text{GLEN}} + O(|\mathcal{N}|)) \\
& + T \cdot O(n_b^2 \cdot d)
\end{aligned} \tag{32}$$

### B.4 Conclusion of Time Complexity

By analyzing Eq.23, Eq.25, and Eq.32, the complexity of CAWN is undoubtedly the largest due to the extensive sampling and anonymization operations. GLEN mainly spends more computation on GCN, TCN, and fusion per time step compared to MP-TGNs. The computation at the time step granularity is much more time-saving than the computation for each specific event. i.e., $|\mathcal{E}| >> T$. Thus, GLEN doesn't spend much more time than MP-TGNs. In addition, the use of TCN brings a substantial increase in efficiency, and the attention-based fusion module of GLEN is also suitable for parallel computing on GPUs. The time complexity of temporal graph approaches depends mainly on the operations related to the number of events because the models focus on each event for processing, and the number of events $|\mathcal{E}|$ is much larger than the number of time steps $T$ or the number of neighbors $|\mathcal{N}|$. Therefore, we can focus mainly on the terms multiplied by $|\mathcal{E}|$, since they are the dominant part of the time complexity. Thus, the time complexity of our method GLEN is mainly due to $|\mathcal{E}| \cdot (\mathcal{C}_{\text{Sampling}}^{\text{GLEN}} + O(|\mathcal{N}|))$. And the time complexities of TGN and CAWN are $|\mathcal{E}| \cdot (\mathcal{C}_{\text{Sampling}}^{\text{TGN}} + O(|\mathcal{N}| \cdot d))$ and $|\mathcal{E}| \cdot O(M \cdot m \cdot (\mathcal{C}_{\text{Sampling}}^{\text{CAWN}} + M))$, respectively. Typically, $M = 64$, $m = 2$, and $d = 172$. So $M \cdot m \cdot M > |\mathcal{N}| \cdot d > |\mathcal{N}|$. Therefore, GLEN achieves competitive efficiency, which can be proved by our experiments in Section 5.4 and Figure 9.

## C Further Details of Experiments

### C.1 Descriptions and Statistics of Datasets

Here we introduce the seven datasets used in the paper.

- **Wikipedia** (Kumar et al., 2019) dataset is a temporal graph containing edits on wiki pages within one month. Users and pages are modeled as nodes, and each timestamped interaction edge represents a user editing a page. Edge features are LIWC-feature vectors (Pennebaker et al., 2001) of edit texts with a length of 172. Dynamic labels of user nodes are available in this dataset, which indicate whether a user is temporarily banned from editing.
- **Reddit** (Kumar et al., 2019) collects the interactions between users and posts on subreddits, and the time span is also one month. Users and subreddit posts are modeled as nodes, and each timestamped interaction edge represents a user's posting request. Edge features are LIWC-feature vectors (Pennebaker et al., 2001) of edit texts with a length of 172. Dynamic labels of user nodes are available in this dataset, which indicate whether a user is temporarily banned from posting.

- **Enron** (Shetty & Adibi, 2004) dataset models emails exchanged among employees of ENRON energy company over three years as a temporal graph.

- **UCI** (Panzarasa et al., 2009) is an unattributed online communication network among students of University of California at Irvine, along with timestamps with a temporal granularity of seconds. The nodes in this dataset represent university students, and temporal edges represent messages posted by the students.

- **UN Trade** (MacDonald et al., 2015) is a temporal food and agriculture trading graph among 181 countries spanning over 30 years. The weight of each edge contained in this dataset is the total sum of normalized agriculture import or export values between two particular countries.

- **MOOC** (Kumar et al., 2019) contains interactions between students and online course content units, where the nodes are the two types of entities. Each edge of the temporal graph represents a student accessing a specific content unit such as problem sets and videos.

- **Flights** (Schäfer et al., 2014) is a dynamic flight network that depicts the evolution of air transportation during the COVID-19 pandemic. Each node represents an airport, and each edge represents a monitored flight. The number of flights between two given airports in a day is specified by the edge weights.

Table 4 gives detailed statistics of the datasets.

Table 4: Summary statistics of the datasets.

| Dataset | Domain | #Nodes | #Events | Attributes for Nodes and Edges | Time Granularity | Duration | Average training time of GLEN per epoch |
|---------|--------|--------|---------|-------------------------------|------------------|----------|------------------------------------------|
| UCI | Social | 1,899 | 59,835 | 0&0 | Unix timestamp | 196 days | 14.10s |
| Enron | Social | 184 | 125,235 | 0&0 | Unix timestamp | 3 years | 23.10s |
| Wikipedia | Social | 9,227 | 157,474 | 172&172 | Unix timestamp | 1 month | 50.81s |
| MOOC | Interaction | 7,144 | 411,749 | 0&4 | Unix timestamp | 17 month | 206.33s |
| UN Trade | Economics | 255 | 507,497 | 172&1 | years | 32 years | 221.85s |
| Reddit | Social | 10,984 | 672,447 | 172&172 | Unix timestamp | 1 month | 324.25s |
| Flights | Transport | 13,169 | 1,927,145 | 172&1 | days | 4 months | 1117.92s |

## C.2 DESCRIPTIONS OF BASELINES

We totally use nine strong baselines in the paper:

- **DyRep (Trivedi et al., 2019):** DyRep uses a two-time scale temporal point process model and parameterizes it with an inductive representation network, which subsequently models the latent mediation process of learning node representations. When an event is observed between two nodes, information flows from the neighborhood of one node to the other and affects the representations of the nodes accordingly.

- **JODIE (Kumar et al., 2019):** JODIE learns embedding trajectories of user nodes and item nodes in temporal interaction networks through two mutually recursive RNNs (Dai et al., 2016; Zhang et al., 2017). Each user or item has a static embedding and a dynamic embedding. The static embedding represents the entity's long-term stationary property, while the dynamic embedding represents the time-varying property. Both embeddings are used to generate the trajectory, which enables Jodie to make predictions from both the stationary and time-varying properties of the user.

- **TGAT (Xu et al., 2020):** TGAT uses the self-attention mechanism (Vaswani et al., 2017b) and introduces a novel functional time encoding technique derived from the Bochner's theorem from classical harmonic analysis (Loomis, 2013) to propagate temporal neighborhood information. The temporal graph attention layer takes the temporal neighborhood with hidden representations (or features) as well as timestamps as input, and the output is the time-aware representation for the target node at any time point.

- **TGN (Rossi et al., 2020):** TGN is a generic inductive framework of deep learning on temporal graph networks. The core modules of TGN include the memory module that memorizes long-term dependencies for each node, the message function that computes messages for the nodes involved in events, the message aggregator that aggregates messages, the memory updater that updates the memories of nodes, and the embedding module that generates temporal node embeddings.

- **CAWN (Wang et al., 2021c):** CAWN extracts temporal random walks and adopts a novel anonymization strategy that replaces node identities with the hitting counts of the nodes based on a set of sampled walks to keep the method inductive and simultaneously establish the correlation between motifs. Causal anonymous walks (Micali & Zhu, 2016) guarantees inductive learning and simultaneously establishes the correlation between motifs (Paranjape et al., 2017; Liu et al., 2021).

- **PINT (Souza et al., 2022):** PINT is a position-encoding injective temporal graph network. PINT defines injective message passing and update steps like MP-TGNs and also augments memory states with novel relative positional features, and these features can replicate all the discriminative benefits available to WA-TGNs.

- **GraphMixer (Cong et al., 2023):** GraphMixer is a conceptually and technically simple architecture for temporal graph learning. It only adopts three simple components: a link-encoder that is only based on MLP to summarize the information from temporal links, a node-encoder that is only based on neighbor mean-pooling to summarize node information, and an MLP-based link classifier that performs link prediction based on the outputs of the encoders. Despite its simplicity, GraphMixer is equipped with fast convergence and impressive generalization ability.

- **TIGER (Zhang et al., 2023):** TIGER focuses on the restarting issue in industrial scenarios and designs a restarter to efficiently generate estimates of current node embeddings using only a small portion of previous data. With the help of the restarter, TIGER can re-initialize the memory warmly at any time such that TIGER can resume training/inference even if our model has been offline for a while. The restarter also enables our proposed methods to run in parallel.

### C.3 IMPLEMENTATION DETAILS OF GLEN

The general default hyper-parameters of GLEN are shown in Table 5. For all attributed and non-attributed datasets, the dimensions of their node features and edge features are fixed as 172. If a dataset lacks features of nodes or edges, zero feature vectors will be assigned to ensure equal sizes of features, which is similar to baselines (Kumar et al., 2019; Xu et al., 2020; Rossi et al., 2020; Souza et al., 2022). For the hyper-parameters investigated in the paper, the optimal combinations of hyper-parameters are shown in Table 6 and the remaining training configurations follow the default values in Table 5.

Table 5: General default hyper-parameters of GLEN.

| Hyper-parameters | Value |
|---|---|
| Batch size | 200 |
| Learning rate | 0.0001 |
| Optimizer | Adam |
| Patience of early stopping strategy | 5 |
| Number of GCN layers | 1 |
| Number of sampled neighbors | 10 |
| Number of attention heads | 4 |
| Dropout ratio | 0.1 |
| Time window size | 2 |
| Number of TCN layers | 3 |
| Kernel size of TCN | 2 |

Table 6: Optimal combinations of GLEN's hyper-parameters for all datasets.

| Hyper-parameters | Wikipedia | Reddit | UCI | Enron | UN Trade | MOOC | Flights |
|---|---|---|---|---|---|---|---|
| Number of sampled neighbors | 10 | 10 | 30 | 20 | 20 | 20 | 20 |
| Time window size | 2 | 2 | 2 | 1 | 1 | 8 | 1 |
| Number of attention heads | 2 | 4 | 1 | 4 | 1 | 4 | 1 |
| Dropout rate | 0.3 | 0.3 | 0.5 | 0.1 | 0.1 | 0.1 | 0.1 |
| Number of TCN layers | 4 | 4 | 3 | 3 | 3 | 3 | 3 |
| Kernel size of TCN | 2 | 3 | 3 | 2 | 2 | 2 | 2 |

## C.4 PSEUDO-CODE OF GLEN

The pseudo-codes of learning our method GLEN are shown in Algorithm 1.

---
**Algorithm 1** Learning algorithm of GLEN
---
**Input:** Set of nodes: $\mathcal{V} = \{1, 2, \ldots, n\}$, observed event stream $\mathcal{E} = \{\mathbf{e}_{uv}(t)\}$, number of network layers $L$, time window size $\Gamma$, number of sampled neighbors $|\mathcal{N}|$, and number of attention heads $\eta$.
**Output:** Embedding $\mathbf{z}_u$ for each node $u$ involved in events.
 1: **for** each batch $b$ **do**
 2:     **for** each event $\mathbf{e}_{uv}(t)$ **do**
 3:         Generate two meassages using Eq.1;
 4:         Update $\mathbf{s}_u(t)$ and $\mathbf{s}_v(t)$ using Eq.4;
 5:     **end for**
 6:     Compose edges of the batch into a graph;
 7:     Normalize the adjacency matrix $\mathbf{A}_b$ to $\widehat{\mathbf{A}}_b$ using Eq.5;
 8:     **for** $l \in L$ **do**
 9:         Perform graph convolution operation on $\mathbf{H}_b^{(l)}$ using Eq.6;
10:     **end for**
11:     Forward $\{\mathbf{H}_{(b-\Gamma+1)}^{(L)}, \mathbf{H}_{(b-\Gamma+2)}^{(L)}, ..., \mathbf{H}_b^{(L)}\}$ into TCN using Eq.7;
12:     Apply dilated convolution operation using Eq.8;
13:     Get $\mathbf{Z}^{\text{Global}}$ using Eq.9;
14:     **for** each node $u$ involved in events of the batch **do**
15:         **for** $l \in L$ **do**
16:             Sample most recent $|\mathcal{N}|$ neighbors $\mathcal{N}_u(t)$ for $u$;
17:             **for** each neighbor node $v \in \mathcal{N}_u(t)$ **do**
18:                 Compute $w_{(v,u,t)}$ using Eq.10;
19:                 Compute $\mathbf{z}_{uv}^{(l)}(t)$ using Eq.11;
20:             **end for**
21:             Perform a weighted sum to obtain $\tilde{\mathbf{h}}_u^{(l)}(t)$ using Eq.12;
22:             Compute $\mathbf{h}_u^{(l)}(t)$ using Eq.13;
23:         **end for**
24:         $\mathbf{z}_u^{\text{Local}} = \mathbf{h}_u^{(L)}(t)$;
25:     **end for**
26:     Get $\mathbf{Z}^{\text{Local}}$ using Eq.14;
27:     **for** $i \in \eta$ **do**
28:         Compute $\mathbf{Q}_i$, $\mathbf{K}_i$, and $\mathbf{V}_i$ using $\mathbf{Z}^{\text{Global}}$, $\mathbf{Z}^{\text{Local}}$, and Eq.15;
29:         Compute $\tilde{\mathbf{Z}}_i$ using Eq.16;
30:     **end for**
31:     Compute $\tilde{\mathbf{Z}}$ using Eq.17;
32:     Compute $\mathbf{Z} = \{\mathbf{z}_u\}$ using Eq.18;
33: **end for**
---

## C.5 DETAILS OF THE EVALUATION PROTOCOL

We conduct evaluation experiments on link prediction and dynamic node classification tasks, as elaborated below.

**Link prediction** (Zhang et al., 2020; Srinivasan & Ribeiro, 2019; Lü & Zhou, 2011) is a fundamental learning task on temporal graphs that focuses on predicting future connections between nodes. The prediction is classified into two categories: inductive and transductive (Xu et al., 2020). In transductive tasks, both node instances of an edge have been observed at training time, and inductive otherwise. As a standard practice used in other works (Xu et al., 2020; Rossi et al., 2020; Wang et al., 2021c), we concatenate the two eigenvectors of each node pair, apply an MLP to obtain the link probability, and then compute the BCE loss. Average precision (AP) is used as the evaluation metric for link prediction. Each method is run 5 times, and the means and standard deviations of the results are taken for comparison. For each batch of data, we perform three benchmark negative sampling strategies (Poursafaei et al., 2022) to sample an equal amount of negative edges to the positive

ones and optimize the BCE loss function. Historical and Inductive strategies are more stringent and challenging evaluation procedures for link prediction specific to temporal graphs. Existing methods have significant performance degradation with the evaluation of both historical and inductive strategies. The three kinds of negative edge sampling strategies based on the official implementation of DGB (Poursafaei et al., 2022) are introduced below as follows.

- **Random Negative Sampling**: To generate negative edges, the random negative sampling procedure simply keeps the timestamps, features, and source nodes of the positive edges while randomly choosing destination nodes from all nodes. This strategy has no collision check between positive and negative instances and lacks the consideration that previously observed edges always reoccur in temporal graphs.

- **Historical Negative Sampling**: Historical negative sampling strategy samples negative edges from the set of edges that have been observed during previous timestamps but are absent in the current step in order to evaluate whether a given method is able to correctly predict an observed training edge would reoccur.

- **Inductive Negative Sampling**: Unlike historical negative sampling, the objective of inductive negative sampling is to evaluate whether a given method can model the reoccurrence pattern of edges only seen during test time. Thus, this strategy samples negative edges from the test instances that were not observed during training and are also absent currently.

The diverse sampling methods are more comprehensive and challenging, allowing for better evaluation and more realistic assessments. Note that if the number of available historical or inductive edges is insufficient to match the number of positive edges, the remaining negative edges are sampled by the random sampling strategy, the same as the benchmark (Poursafaei et al., 2022). The key way to improve link prediction performance is optimizing the node embeddings so that nodes tending to interact with each other as the graph evolves have similar embeddings.

**Dynamic node classification** (Aggarwal & Li, 2011; Xu et al., 2019) task is to predict the labels of nodes that may change over time. Publicly available datasets for node classification in the temporal graphs are rare, so we only use three datasets (Wikipedia, Reddit, and MOOC) for demonstration. We also use an MLP as the decoder that maps the node embedding to the class probability. Due to the skew of label distribution, we employ the area under receiver operating characteristic (ROC AUC) as the main performance metric. All these settings mentioned above are consistent with baselines (Kumar et al., 2019; Xu et al., 2020; Rossi et al., 2020).

## C.6   DETAILS OF THE ABLATION STUDY

- **w/o GCN**: We remove the GCN (Kipf & Welling, 2016) in the global embedding module of GLEN and denote this variant as **w/o GCN**. In this case, the memories of nodes are not processed by GCN but are directly input into TCN.

- **w/o TCN**: We remove the TCN (Bai et al., 2018) in the global embedding module of GLEN and denote this variant as **w/o TCN**. In this case, GLEN only utilizes the output of GCN at the current time step as the global embeddings of nodes. The GCN outputs of the last few time steps are no longer input into TCN.

- **w/o Global**: We remove the entire global embedding module of GLEN and denote this variant as **w/o Global**. In this case, the memories of nodes rather than global embeddings are directly treated as the 'query' of the attention (Vaswani et al., 2017a) mechanism in the cross-perspective fusion module.

- **w/o Local**: We remove the entire local embedding module of GLEN and denote this variant as **w/o Local**. In this case, the memories of nodes rather than local embeddings are directly treated as the 'key' and 'value' of the attention (Vaswani et al., 2017a) mechanism in the cross-perspective fusion module.

- **w/o Fusion**: We remove the cross-perspective fusion module of GLEN and denote this variant as **w/o Fusion**. In this case, the global and local embeddings of the same node are fused in summation form to get the final node representation. The degradation of GLEN performance when removing the cross-perspective fusion module in our ablation study is somewhat indicative

of the effectiveness of this module since it takes into account the high-order information in the graph.

# D  SUPPLEMENTARY RESULTS

Due to the limitation of space, we report the results of additional experiments in Table 7, Figure 9, Figure 10, and Figure 11 respectively.

## D.1  SUPPLEMENTARY EXPERIMENTAL RESULTS ON TRANSDUCTIVE LINK PREDICTION

We report the Average Precision (AP) in transductive link prediction under the evaluation of three negative sampling strategies in Table 7. GLEN also achieves state-of-the-art (SOTA) performance.

## D.2  SUPPLEMENTARY EXPERIMENTAL RESULTS OF EFFICIENCY

We further evaluate the trade-off capability between performance and training efficiency for various models on Wikipedia and MOOC datasets, which is shown in Figure 9. Note that the training time of PINT here does not include precomputing the positional features, otherwise the training time of PINT will be even much longer than CAWN. Since PINT takes too long on the largest Flights dataset, we did not include PINT's results on Flights in our experimental results in Table 2 and Table 7. The results further prove the impressive balance between inference precision and training speed of GLEN.

## D.3  SUPPLEMENTARY EXPERIMENTAL RESULTS FOR THE INFLUENCE OF HYPER-PARAMETERS

Additional investigation of the time window size $\Gamma$ is shown in Figure 10. The performance of GLEN drops as the time window size increases, which demonstrates that the most recent information is sufficient for GLEN to generate ideal node embeddings. Large $\Gamma$ may lead to stale information and performance degradation.

Additional investigation of the number of sampled neighbors $|\mathcal{N}|$ is shown in Figure 11. An interesting insight is that the performance of GLEN on several datasets is scarcely affected by the number of sampled neighbors. However, other TGNs techniques typically require a certain number (usually 10 or 20) of neighbor nodes to achieve their best performanceWang et al. (2021c); Rossi et al. (2020). This confirms, to some extent, that global embeddings bring complementary information to local embeddings through the fusion module of GLEN.

## D.4  SUPPLEMENTARY EXPERIMENTAL RESULTS OF ABLATION STUDY

The results of the ablation study on dynamic node classification are summarized in Table 8.

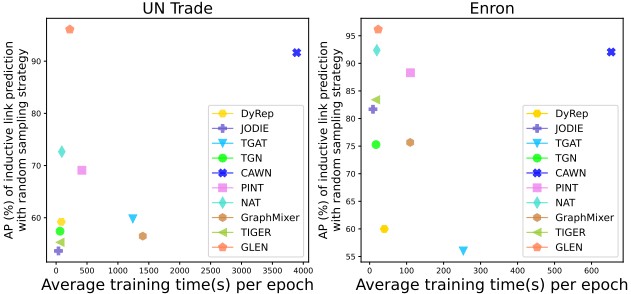

Figure 9: Trade-off between AP (%) in link prediction and training time on Wikipedia and Enron datasets. The reported AP (over 5 runs) is computed with the random negative sampling strategy as well as the inductive setting in percentage format. The methods closer to the upper left corner of the figure are more ideal.

Table 7: Average Precision (%) of link prediction under different negative sampling strategies in the transductive setting (over 5 runs). (**First** second)

| NS Strategy | Methods | Wikipedia | Reddit | Enron | UCI | UN Trade | MOOC | Flights |
|---|---|---|---|---|---|---|---|---|
| Random | DyRep | 69.37±0.82 | 56.77±1.36 | 62.35±2.86 | 65.86±3.34 | 60.00±1.70 | 60.86±0.28 | 95.22±0.42 |
| | JODIE | 98.31±0.07 | 99.86±0.03 | 87.05±0.56 | 98.42±0.24 | 64.31±1.48 | 82.17±5.45 | 96.63±0.55 |
| | TGAT | 95.57±0.13 | 98.23±0.05 | 59.03±2.98 | 77.32±1.10 | 60.48±1.13 | 70.72±0.45 | 94.17±0.04 |
| | TGN | 97.89±0.17 | 96.28±0.56 | 80.97±1.83 | 88.35±3.59 | 65.89±0.60 | 84.31±2.76 | 91.24±0.42 |
| | CAWN | 99.56±0.27 | 99.74±0.24 | 90.80±1.48 | 98.45±1.03 | 91.61±0.50 | 87.44±2.31 | 99.59±0.08 |
| | PINT | 98.16±0.16 | 98.51±0.10 | 92.83±0.06 | 97.20±0.15 | 65.14±0.61 | 77.97±1.93 | - |
| | GraphMixer | 96.99±0.09 | 97.23±0.02 | 81.86±0.58 | 92.91±0.47 | 57.26±2.62 | 82.40±0.54 | 90.99±0.04 |
| | TIGER | 98.62±1.10 | 99.04±1.18 | 84.35±0.39 | 93.07±0.41 | 56.98±0.19 | 84.05±0.72 | 94.15±0.53 |
| | **GLEN** | **99.99±0.01** | **99.99±0.01** | **93.67±0.48** | **99.55±0.90** | **97.83±0.15** | **98.39±2.52** | **99.96±0.01** |
| Historical | DyRep | 41.72±0.46 | 44.97±1.08 | 56.95±4.77 | 48.44±1.13 | 58.48±2.545 | 41.22±0.43 | 67.32±1.02 |
| | JODIE | 42.82±0.20 | 46.70±0.65 | 43.76±0.11 | 44.98±3.24 | 63.11±1.24 | 33.50±0.10 | 65.97±1.49 |
| | TGAT | 76.09±0.28 | 77.74±0.57 | 53.13±2.46 | 60.25±0.69 | 51.55±2.83 | 63.99±2.13 | 72.29±0.22 |
| | TGN | 78.47±2.33 | 73.08±2.07 | 70.09±1.88 | 73.69±4.60 | 60.90±1.03 | 62.98±4.51 | 63.86±1.47 |
| | CAWN | 80.31±9.61 | 80.08±0.75 | 58.58±6.46 | 81.14±0.94 | 86.96±0.62 | 96.87±4.91 | 51.83±0.19 |
| | PINT | 71.36±6.58 | 77.80±6.70 | 84.16±0.52 | 94.44±0.46 | 59.08±1.73 | 72.63±3.15 | - |
| | GraphMixer | 90.83±0.25 | 77.96±0.40 | 78.89±0.96 | 85.63±0.35 | 59.64±2.71 | 77.56±1.05 | 71.55±0.23 |
| | **GLEN** | **97.67±0.16** | **98.74±1.25** | **94.35±2.24** | **98.24±1.13** | **97.65±0.08** | **99.89±0.11** | **85.46±0.39** |
| Inductive | DyRep | 63.57±1.24 | 51.63±2.51 | 57.55±3.43 | 52.51±0.61 | 60.25±0.85 | 57.96±1.32 | 70.76±1.00 |
| | JODIE | 40.95±0.29 | 47.21±0.61 | 47.18±0.20 | 54.11±2.28 | 64.85±1.25 | 42.68±4.12 | 67.74±0.39 |
| | TGAT | 81.76±0.22 | 88.28±0.40 | 56.83±5.05 | 61.62±0.49 | 54.67±3.59 | 56.85±0.41 | 75.35±0.10 |
| | TGN | 82.76±1.07 | 82.77±0.72 | 68.23±1.55 | 70.69±0.73 | 64.50±1.52 | 58.64±4.17 | 64.84±1.55 |
| | CAWN | 70.53±1.52 | 78.31±1.76 | 61.81±3.97 | 80.04±6.61 | 89.41±1.05 | 91.26±1.32 | 53.55±0.17 |
| | PINT | 65.49±7.09 | 83.10±0.85 | 78.39±0.52 | 86.72±0.52 | 62.83±2.61 | 71.28±3.10 | - |
| | GraphMixer | 88.43±0.40 | 85.04±0.17 | 75.14±0.93 | 77.97±0.31 | 62.86±3.21 | 74.34±0.38 | 74.84±0.20 |
| | **GLEN** | **95.20±0.36** | **99.24±0.82** | **94.99±1.91** | **93.30±3.31** | **97.45±2.11** | **99.86±0.19** | **89.83±0.77** |

Table 8: Average ROC AUC (%) for ablation study of GLEN in dynamic node classification.

| Ablation | Wikipedia | Reddit | MOOC |
|---|---|---|---|
| w/o GCN | 87.81 | 63.77 | 67.00 |
| w/o TCN | 86.30 | 65.62 | 66.71 |
| w/o Global | 86.74 | 67.08 | 61.79 |
| w/o Local | 87.14 | 61.29 | 69.00 |
| w/o Fusion | 87.80 | 59.26 | 58.90 |
| **GLEN** | **90.16** | **70.21** | **71.49** |

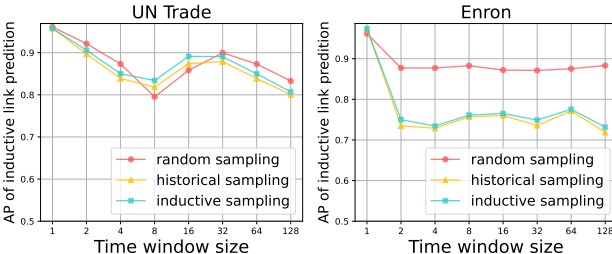

Figure 10: Performance of GLEN with different time window sizes $\Gamma$ in the global embedding module on UN Trade and Enron datasets. The AP (over 5 runs) in the inductive link prediction task is reported.

# E   DISCUSSION OF SCALABILITY

Since link prediction is the main task of temporal graph learning and there are few datasets suitable for node classification, we focus mainly on link prediction like other works (Wang et al., 2021c; Poursafaei et al., 2022). In terms of scalability, our approach actually has potential scalability. Firstly, scalability of GLEN can be improved by replacing the graph neural network in the global embedding module with an advanced method suitable for large graphs, such as GraphSAGE (Hamilton et al., 2017a), or by using distributed training, etc. Secondly, the datasets used in our paper already include the largest dataset (Flights (Schäfer et al., 2014)) of the publicly available benchmark (Poursafaei et al., 2022). The main focus of our paper is to further improve the model performance since the presentation of the new benchmark (Poursafaei et al., 2022) has made the

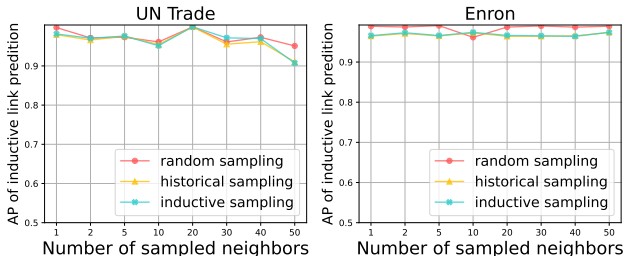

Figure 11: Performance of GLEN with different numbers of sampled neighbors$|\mathcal{N}|$ in the local embedding module on UN Trade and Enron datasets. The reported AP (over 5 runs) in the inductive link prediction task is reported.

prediction precision of existing methods face unprecedented challenges. Especially for baselines, there is a significant decrease in performance under the evaluation of historical and inductive negative sampling strategies which prove to reflect real-world considerations for temporal graphs. As for industry-level temporal graph methods, there are few related works. Thirdly, the "largeness" of temporal graph datasets is mainly reflected in the number of edges (Wang et al., 2021c) since temporal graph datasets are modeled as event streams. Existing models also focus on processing each event. The table of dataset statistics we provided in Table 4 is organized according to the training time on the dataset from smallest to largest. It can be seen that datasets with more edges take more time, and there is no correlation with the number of nodes. So the numbers of nodes in the widely used datasets are limited, and the number of events determines the training overhead (Wang et al., 2021c). Currently, due to the new benchmark (Poursafaei et al., 2022) being proposed, the prediction precision of temporal graph methods is facing great challenges. So our work focuses more on further improving the model performance. Scalability seems to be orthogonal to our work, we will try it in our future work.

