# OpenReview forum: "Boosting Temporal Graph Learning From Global and Local Perspectives"
_ICLR.cc/2024/Conference — Submitted to ICLR 2024_

### Official Review · Reviewer_u41A · 2023-10-29

**Soundness:** 2 fair
**Presentation:** 2 fair
**Contribution:** 2 fair
**Rating:** 5
**Confidence:** 4

**Summary:**

This paper focuses on temporal graph representation learning. Different from exiting works that utilize GNN-RNN frameworks or local subgraph information to learn effective node representations, this paper proposes to generate node embeddings by considering both global and local perspectives. Specifically, GCN-TCN is utilized to encode global graph information; TGN is used to model local graph information; Then, self-attention mechanism is employed to aggregate node representations from both global and local embeddings. Experimental results on seven datasets demonstrate that the proposed model can achieve satisfied performance on both link prediction and node classification tasks.

**Strengths:**

1)	This paper investigates temporal graph representation learning, which is an important topic in the graph community.
2)	Various ablation studies are given to show the effectiveness of the proposed components.
3)	Time complexity analysis is given, and model efficiency analyses are also presented in the experiment section.

**Weaknesses:**

1)	The technical contribution of this paper is quite limited. The three components in this paper are all from existing works. The global module is from TCN, and the local module is from TGN. The cross-perspective fusion is a self-attention module. No new module is proposed.
2)	In the cross-perspective fusion module, it is not clear why query $Q_i$ is the linear projection of $Z^{Global}$. What if we use $Z^{local}$ to generate $Q_i$? In this case, $\tilde{z}$ is the weighted combination of $Z^{global}$ and $z_u = FFN(\tilde{z}_u \Vert z_u^{local})$. More ablation studies should be conducted.
3)	In Table 2, it is not clear why the performance of GraphMixer and TIGER are missing in the setting of historical and inductive negative sampling.
4)	There are lots of other manners to combine both the global and local perspectives. For instance, $z_u$ is directly generated by concatenating $Z^{local}$ and $Z^{global}$. $z_u$ can also be aggregated with simple attention mechanism instead of self-attention.
5)	In Equation 10, what if some of the nodes do not have $|\mathcal{N}|$ neighbors? Are there any strategies to handle this situation?

There are lots of typos in the paper. For example,

In Figure 1, “whether nodes B and C will interact at t3” should be “whether nodes B and D will interact at t3”

“we focus on further improve the model performance” should be “we focus on further improving the model performance”

“Wheras, GLEN can keep” should be “Whereas, GLEN can keep”

**Questions:**

1)	In Table 2, it is not clear why the performance of GraphMixer and TIGER are missing in the setting of historical and inductive negative sampling.
2)	There are lots of other manners to combine both the global and local perspectives. For instance, $z_u$ is directly generated by concatenating $Z^{local}$ and $Z^{global}$. $z_u$ can also be aggregated with simple attention mechanism instead of self-attention.
3)	In Equation 10, what if some of the nodes do not have $|\mathcal{N}|$ neighbors? Are there any strategies to handle this situation?

---

> ### Author Response · Authors · 2023-11-14
> **Part 1 for Reviewer u41A**
>
> Thanks a lot for your insightful comments and detailed suggestions, which are very helpful for us to further improve this paper. We would like to thank you for bringing up the typos in our paper. We have corrected them and further improved our paper. Next, we'll answer your questions one by one. We hope our following answers will address the points you have raised and improve your view of our work.
>
> **W1:**
>
> Thanks for your comments.
>
> To the best of our knowledge, we are the first in the field of temporal graph learning to propose a method that simultaneously models the graph structure from an entire global perspective and a local subgraph perspective, and fuses all node embeddings across views. Our method fills the gap in existing methods that only focus on one perspective and highlights the importance of considering both views. Our concept and related analysis of the idea that modeling temporal graphs from both global and local perspectives is novel and has research significance in this field. And we propose a completely new model based on this motivation.
>
> Therefore, our novelty lies in our contributed insights, motivation as well as the innovatively proposed method. The novelty of each module is specified below:
>
> - The novelty of the global embedding module is that there is no work that combines GNN and TCN for learning on temporal graphs to obtain more efficient and effective results, to the best of our knowledge.
> - In the local embedding module, we designed a new time interval weighting algorithm that is more concise and efficient, which is different from TGN.
> - The Cross-Perspective Fusion Module is designed to better fuse global and local embeddings, and its novelty lies in the motivation for integration based on the semantic correlation among all node embeddings, while the attention mechanism is just a way for us to implement the module.
>
> It is also of research significance to construct effective and efficient methods that work well in experiments and applications using basic, simple components. This inspires us that it may not be necessary to adopt a complex approach to temporal graph learning. Often basic models with novel and solid insights can also make good gains and achieve a better balance between inference precision and efficiency.
>
> We hope our answers address your concerns.
>
>
>
> **W2:**
>
> Thanks for pointing this out. Due to space constraints, we did not explain this in the paper. We use the global embedding as the query and the local embedding as the key and value for 2 main reasons:
>
> - In terms of modeling temporal information, local embeddings are obtained by encoding specific timestamp information, whereas global module relies more on evolution patterns across time slices by modeling sequencial information at a coarse-grained level using TCN. Therefore, in order to better utilize the fine-grained timestamp information, the hidden state of local embeddings need to be utilized.
>
> - In terms of topological information, although both local and global modules aggregate neighborhood information for nodes, the local module can aggregate neighbor information more effectively than the graph convolution of the global module due to the use of our devised  weighted sum algorithm based on time interval. And as we mentioned in the introduction, local and global are oriented to graph structures with different degrees of integrity (local subgraphs or entire graph snapshots), so the operations of the local module are more fine-grained.
>
> To better validate the soundness of our considerations and address your concern, we further compare the performance (Average Precision, %) of the model before and after (denoted as GLEN-swap) swapping $Z^{Global}$ and $Z^{Local}$ in the attention, as shown in the following tables:

---

> ### Author Response · Authors · 2023-11-14
> **Part 2 for Reviewer u41A**
>
> | Dataset                      |           | Wikipedia  | Wikipedia      | Wikipedia     | Reddit     | Reddit         | Reddit        | Enron      | Enron          | Enron         | UCI        | UCI            | UCI           |
> | ---------------------------- | --------- | ---------- | -------------- | ------------- | ---------- | -------------- | ------------- | ---------- | -------------- | ------------- | ---------- | -------------- | ------------- |
> | **NS Strategy**              |           | **Random** | **Historical** | **Inductive** | **Random** | **Historical** | **Inductive** | **Random** | **Historical** | **Inductive** | **Random** | **Historical** | **Inductive** |
> | Transductive link prediction | GLEN      | 99.99±0.01 | 97.67±0.16     | 95.20±0.36    | 99.99±0.01 | 98.74±1.25     | 99.24±0.82    | 93.67±0.48 | 94.35±2.24     | 94.99±1.91    | 99.55±0.90 | 98.24±1.13     | 93.30±3.31    |
> | Transductive link prediction | GLEN-swap | 96.79±0.69 | 89.97±3.45     | 84.38±11.43   | 98.19±0.10 | 86.60±2.23     | 85.13±6.27    | 88.46±0.84 | 77.79±6.31     | 71.76±2.56    | 82.56±5.08 | 77.36±1.13     | 76.80±0.62    |
> |                              | Reduction | ⬇3.2%      | ⬇7.88%         | ⬇11.37%       | ⬇1.8%      | ⬇12.29%        | ⬇14.20%       | ⬇4.86%     | ⬇17.55%        | ⬇24.46%       | ⬇17.07%    | ⬇31.43%        | ⬇28.4%        |
> | Inductive link prediction    | GLEN      | 99.95±0.05 | 96.25±0.27     | 96.13±0.29    | 99.85±0.28 | 97.31±2.46     | 97.28±2.48    | 96.15±1.61 | 97.28±0.53     | 97.38±0.46    | 99.11±0.30 | 95.94±2.00     | 95.43±2.63    |
> | Inductive link prediction    | GLEN-swap | 96.82±0.46 | 85.72±11.98    | 85.86±11.81   | 96.18±0.17 | 79.84±1.07     | 81.81±2.90    | 88.48±2.68 | 62.86±3.14     | 63.52±3.14    | 89.91±1.99 | 78.98±0.83     | 79.03±0.65    |
> |                              | Reduction | ⬇3.13%     | ⬇10.94%        | ⬇10.68%       | ⬇3.68%     | ⬇17.93%        | ⬇15.9%        | ⬇18.38%    | ⬇35.38%        | ⬇34.77%       | ⬇19.37%    | ⬇28.1%         | ⬇27.66%       |
>
> | Dataset                      |           | UN Trade    | UN Trade       | UN Trade      | MOOC       | MOOC           | MOOC          | Flights    | Flights        | Flights       |
> | ---------------------------- | --------- | ----------- | -------------- | ------------- | ---------- | -------------- | ------------- | ---------- | -------------- | ------------- |
> | **NS Strategy**              |           | **Random**  | **Historical** | **Inductive** | **Random** | **Historical** | **Inductive** | **Random** | **Historical** | **Inductive** |
> | Transductive link prediction | GLEN      | 97.83±0.15  | 97.65±0.08     | 97.45±2.11    | 98.39±2.52 | 99.89±0.11     | 99.86±0.19    | 99.96±0.01 | 85.46±0.39     | 89.83±0.77    |
> | Transductive link prediction | GLEN-swap | 83.43±11.32 | 72.62±10.57    | 72.72±10.48   | 97.87±3.92 | 85.26±3.49     | 88.27±5.24    | 78.88±3.92 | 79.92±5.33     | 79.61±14.67   |
> |                              | Reduction | ⬇14.72%     | ⬇25.63%        | ⬇25.38%       | ⬇0.53%     | ⬇14.65%        | ⬇11.61%       | ⬇21.09%    | ⬇6.48%         | ⬇11.49%       |
> | Inductive link prediction    | GLEN      | 96.09±0.12  | 95.78±2.32     | 95.76±2.32    | 96.48±4.02 | 99.53±0.93     | 99.54±0.91    | 99.36±0.17 | 76.96±0.54     | 77.23±0.61    |
> | Inductive link prediction    | GLEN-swap | 80.16±8.71  | 69.59±7.56     | 69.58±7.51    | 95.50±3.54 | 85.98±3.84     | 85.88±3.69    | 77.50±3.54 | 75.43±4.39     | 75.58±5.25    |
> |                              | Reduction | ⬇16.58%     | ⬇27.34%        | ⬇27.34%       | ⬇1.02%     | ⬇13.61%        | ⬇13.72%       | ⬇22%       | ⬇1.99%         | ⬇2.14%        |
>
> The performance decreases on almost all datasets after swapping $Z^{Global}$ and $Z^{Local}$ . The experimental results somewhat justify our setting of the query, key and value.

---

> ### Author Response · Authors · 2023-11-14
> **Part 3 for Reviewer u41A**
>
> **W3&Q1:**
>
> Thanks for pointing this out. Since these two methods were presented after the benchmark [1] that proposes the historical and inductive negative sampling, and due to the coupling and complex frameworks of the codes, it is difficult to evaluate these two methods using historical and inductive negative sampling.
>
> When we submitted this paper, we did not have time to successfully apply both strategies to these methods. However, we have now been able to evaluate GraphMixer with both strategies and the results are presented in the table below. We have updated the results in our paper.
>
> | NS  Strategy | Task         | Method     | Wikipedia      | Reddit         | Enron          | UCI            | Untrade        | MOOC           | Flights        |
> | ------------ | ------------ | ---------- | -------------- | -------------- | -------------- | -------------- | -------------- | -------------- | -------------- |
> | Historical   | Transductive | GraphMixer | 90.83±0.25     | 77.96±0.40     | 78.89±0.96     | 85.63±0.35     | 59.64±2.71     | 77.56±1.05     | 71.55±0.23     |
> | Historical   | Transductive | GLEN       | **97.67±0.16** | **98.74±1.25** | **94.35±2.24** | **98.24±1.13** | **97.65±0.08** | **99.89±0.11** | **85.46±0.39** |
> | Historical   | Inductive    | GraphMixer | 88.02±0.39     | 64.48±0.36     | 73.18±1.20     | 80.29±0.31     | 58.92±2.67     | 74.07±0.73     | 65.23±0.23     |
> | Historical   | Inductive    | GLEN       | **96.25±0.27** | **97.31±2.46** | **97.28±0.53** | **95.94±2.00** | **95.78±2.32** | **99.53±0.93** | **76.96±0.54** |
> | Inductive    | Transductive | GraphMixer | 88.43±0.40     | 85.04±0.17     | 75.14±0.93     | 77.97±0.31     | 62.86±3.21     | 74.34±0.38     | 74.84±0.20     |
> | Inductive    | Transductive | GLEN       | **95.20±0.36** | **99.24±0.82** | **94.99±1.91** | **93.30±3.31** | **97.45±2.11** | **99.86±0.19** | **89.83±0.77** |
> | Inductive    | Inductive    | GraphMixer | 83.91±0.54     | 63.96±0.26     | 73.19±1.19     | 80.33±0.31     | 58.89±2.66     | 74.08±0.73     | 63.13±0.15     |
> | Inductive    | Inductive    | GLEN       | **96.13±0.29** | **97.28±2.48** | **97.38±0.46** | **95.43±2.63** | **95.76±2.32** | **99.54±0.91** | **77.23±0.61** |
>
> Implementing historical and inductive strategies requires recording the historical interaction edges of all nodes and sampling accordingly during the testing phase. Since TIGER focuses primarily on model restarting and designs a restarter to output the warm initialization for node memory, the memory is very different from the other baselines. The restarter re-initializes the memory using only a small portion of the recent events. Therefore, the historical and inductive negative sampling strategies are not applicable to TIGER. We realize that this may be confusing to readers who are less familiar with the field. We will add relevant explanations in the future version of our paper.
>
> [1] Poursafaei, Farimah , et al. "Towards Better Evaluation for Dynamic Link Prediction." (NIPS, 2022). URL https://openreview.net/forum?id=1GVpwr2Tfdg.

---

> ### Author Response · Authors · 2023-11-14
> **Part 4 for Reviewer u41A**
>
> **W4&Q2:**
>
> What we used in the cross-perspective fusion module is not self-attention. As for other methods of combination, as we said in Appendix C.6, the **w/o. Fusion** variant in the ablation experiments is to simply add $Z^{local}$ and $Z^{local}$, which leads to degradation in performance. As for the concatenation approach and simple attention mechanism (We think you may be referring to weighted summation using learned weights.) you mentioned, we add experiments to verify that the module we designed is more effective.
>
> **Experimental results of GLEN using concatenation for fusion:**
>
> | Dataset                      |                          | Wikipedia  |   Wikipedia    |   Wikipedia   |   Reddit   |     Reddit     |    Reddit     |   Enron    |     Enron      |     Enron     |    UCI     |      UCI       |      UCI      |
> | ---------------------------- | ------------------------ | :--------: | :------------: | :-----------: | :--------: | :------------: | :-----------: | :--------: | :------------: | :-----------: | :--------: | :------------: | :-----------: |
> | **NS Strategy**              |                          | **Random** | **Historical** | **Inductive** | **Random** | **Historical** | **Inductive** | **Random** | **Historical** | **Inductive** | **Random** | **Historical** | **Inductive** |
> | Transductive link prediction | GLEN                     | 99.99±0.01 |   97.67±0.16   |  95.20±0.36   | 99.99±0.01 |   98.74±1.25   |  99.24±0.82   | 93.67±0.48 |   94.35±2.24   |  94.99±1.91   | 99.55±0.90 |   98.24±1.13   |  93.30±3.31   |
> |                              | GLEN-using concatenation | 97.13±0.23 |   71.10±2.06   |  77.14±4.42   | 96.35±0.25 |   68.28±0.65   |  62.25±0.54   | 79.11±5.48 |   66.95±2.22   |  65.45±3.68   | 85.34±2.99 |   69.06±2.66   |  68.71±1.08   |
> |                              | Reduction                |   ⬇2.86%   |     ⬇27.2%     |    ⬇18.97%    |   ⬇3.64%   |    ⬇30.85%     |    ⬇37.27%    |  ⬇15.54%   |    ⬇29.04%     |    ⬇31.1%     |  ⬇14.27%   |     ⬇29.7%     |    ⬇26.36%    |
> | Inductive link prediction    | GLEN                     | 99.95±0.05 |   96.25±0.27   |  96.13±0.29   | 99.85±0.28 |   97.31±2.46   |  97.28±2.48   | 96.15±1.61 |   97.28±0.53   |  97.38±0.46   | 99.11±0.30 |   95.94±2.00   |  95.43±2.63   |
> |                              | GLEN-using concatenation | 96.61±0.18 |   77.48±4.04   |  77.27±3.61   | 96.14±0.14 |   59.10±0.39   |   59.29±051   | 75.85±7.33 |   62.24±2.83   |  62.31±2.83   | 81.42±1.33 |   70.98±0.70   |  71.15±0.65   |
> |                              | Reduction                |   ⬇3.34%   |     ⬇19.5%     |    ⬇19.62%    |   ⬇3.72%   |    ⬇39.27%     |    ⬇37.99%    |  ⬇21.11%   |    ⬇36.02%     |    ⬇36.01%    |  ⬇17.85%   |    ⬇26.02%     |    ⬇25.44%    |
>
> | Dataset                      |                          |  UN Trade  |    UN Trade    | UN Trade      |    MOOC    |      MOOC      | MOOC          | Flights    | Flights        | Flights       |
> | ---------------------------- | ------------------------ | :--------: | :------------: | ------------- | :--------: | :------------: | ------------- | ---------- | -------------- | ------------- |
> | **NS Strategy**              |                          | **Random** | **Historical** | **Inductive** | **Random** | **Historical** | **Inductive** | **Random** | **Historical** | **Inductive** |
> | Transductive link prediction | GLEN                     | 97.83±0.15 |   97.65±0.08   | 97.45±2.11    | 98.39±2.52 |   99.89±0.11   | 99.86±0.19    | 99.96±0.01 | 85.46±0.39     | 89.83±0.77    |
> |                              | GLEN-using concatenation | 94.97±0.06 |   89.67±0.21   | 89.96±0.03    | 82.04±3.06 |   68.31±4.95   | 73.63±1.27    | 92.03±0.52 | 68.28±0.65     | 65.62±2.43    |
> |                              | Reduction                |   ⬇2.92%   |     ⬇8.17%     | ⬇7.69%        |  ⬇16.62%   |    ⬇31.61%     | ⬇26.27%       | ⬇7.93%     | ⬇20.1%         | ⬇26.95%       |
> | Inductive link prediction    | GLEN                     | 96.09±0.12 |   95.78±2.32   | 95.76±2.32    | 96.48±4.02 |   99.53±0.93   | 99.54±0.91    | 99.36±0.17 | 76.96±0.54     | 77.23±0.61    |
> |                              | GLEN-using concatenation | 92.87±0.01 |   89.66±0.05   | 89.68±0.14    | 81.20±3.97 |   72.20±2.64   | 72.08±2.95    | 84.50±0.52 | 59.19±0.39     | 59.32±1.91    |
> |                              | Reduction                |   ⬇3.35%   |     ⬇6.39%     | ⬇6.35%        |  ⬇15.84%   |    ⬇27.46%     | ⬇27.59%       | ⬇14.96%    | ⬇23.09%        | ⬇23.19%       |

---

> ### Author Response · Authors · 2023-11-14
> **Part 5 for Reviewer u41A**
>
> **Experimental results of GLEN using simple attention mechanism for fusion:**
>
> | Dataset                      |                             | Wikipedia  | Wikipedia  | Wikipedia  |   Reddit   |   Reddit   |   Reddit   |   Enron    |   Enron    |   Enron    |    UCI     |    UCI     |    UCI     |
> | ---------------------------- | --------------------------- | :--------: | :--------: | :--------: | :--------: | :--------: | :--------: | :--------: | :--------: | :--------: | :--------: | :--------: | :--------: |
> | NS Strategy                  |                             |   Random   | Historical | Inductive  |   Random   | Historical | Inductive  |   Random   | Historical | Inductive  |   Random   | Historical | Inductive  |
> | Transductive link prediction | GLEN                        | 99.99±0.01 | 97.67±0.16 | 95.20±0.36 | 99.99±0.01 | 98.74±1.25 | 99.24±0.82 | 93.67±0.48 | 94.35±2.24 | 94.99±1.91 | 99.55±0.90 | 98.24±1.13 | 93.30±3.31 |
> |                              | GLEN-using simple attention | 95.04±0.24 | 77.20±3.41 | 83.26±0.30 | 98.77±0.23 | 62.20±2.45 | 64.81±2.23 | 92.62±0.66 | 94.24±0.85 | 93.47±0.74 | 85.19±0.78 | 68.60±0.71 | 68.72±0.95 |
> |                              | Reduction                   |   ⬇4.95%   |  ⬇20.96%   |  ⬇12.54%   |   ⬇1.22%   |  ⬇37.01%   |  ⬇34.69%   |   ⬇1.12%   |   ⬇0.12%   |   ⬇1.60%   |  ⬇14.42%   |  ⬇30.17%   |  ⬇26.35%   |
> | Inductive link prediction    | GLEN                        | 99.95±0.05 | 96.25±0.27 | 96.13±0.29 | 99.85±0.28 | 97.31±2.46 | 97.28±2.48 | 96.15±1.61 | 97.28±0.53 | 97.38±0.46 | 99.11±0.30 | 95.94±2.00 | 95.43±2.63 |
> |                              | GLEN-using simple attention | 95.68±0.08 | 84.12±0.55 | 84.40±0.48 | 95.17±0.39 | 68.10±0.88 | 68.13±0.90 | 95.20±0.13 | 94.63±1.78 | 94.76±1.58 | 81.03±0.67 | 70.91±0.88 | 70.97±0.98 |
> |                              | Reduction                   |   ⬇4.27%   |  ⬇12.60%   |   ⬇12.2%   |   ⬇4.69%   |  ⬇30.02%   |  ⬇29.97%   |   ⬇0.98%   |   ⬇2.72%   |   ⬇2.69%   |  ⬇18.24%   |  ⬇26.09%   |  ⬇25.63%   |
>
> | Dataset                      |                             |  UN Trade  |  UN Trade  |  UN Trade  |    MOOC    |    MOOC    |    MOOC    |  Flights   |  Flights   |  Flights   |
> | ---------------------------- | --------------------------- | :--------: | :--------: | :--------: | :--------: | :--------: | :--------: | :--------: | :--------: | :--------: |
> | NS Strategy                  |                             |   Random   | Historical | Inductive  |   Random   | Historical | Inductive  |   Random   | Historical | Inductive  |
> | Transductive link prediction | GLEN                        | 97.83±0.15 | 97.65±0.08 | 97.45±2.11 | 98.39±2.52 | 99.89±0.11 | 99.86±0.19 | 99.96±0.01 | 85.46±0.39 | 89.83±0.77 |
> |                              | GLEN-using simple attention | 69.87±0.29 | 62.44±0.38 | 66.17±0.98 | 76.33±3.65 | 66.46±4.31 | 69.81±2.18 | 91.63±1.06 | 66.65±0.79 | 67.78±0.58 |
> |                              | Reduction                   |  ⬇28.58%   |  ⬇36.06%   |  ⬇32.10%   |  ⬇22.42%   |  ⬇33.47%   |  ⬇30.09%   |   ⬇8.33%   |  ⬇22.01%   |  ⬇24.55%   |
> | Inductive link prediction    | GLEN                        | 96.09±0.12 | 95.78±2.32 | 95.76±2.32 | 96.48±4.02 | 99.53±0.93 | 99.54±0.91 | 99.36±0.17 | 76.96±0.54 | 77.23±0.61 |
> |                              | GLEN-using simple attention | 63.67±0.64 | 55.13±0.07 | 55.18±0.25 | 72.64±4.69 | 65.26±3.40 | 65.10±3.26 | 85.71±1.60 | 60.95±1.10 | 60.82±1.14 |
> |                              | Reduction                   |  ⬇33.74%   |  ⬇42.44%   |  ⬇42.38%   |  ⬇24.71%   |  ⬇34.43%   |  ⬇34.60%   |  ⬇13.74%   |   20.80%   |  ⬇21.34%   |
>
> Actually, our main motivations for proposing the use of attention in the cross-perspective fusion module are the following points:
>
> - To better capture the semantic correlations between global embeddings and local embeddings of all nodes, which cannot be achieved by simple addition or concatenation.
> - The use of attentional aggregation allows the consideration of the eigenvectors of all nodes jointly, whereas addition or concatenation only operates on the global and local embeddings of the same node.
>
> As we mentioned in Appendix A, global and local embeddings can capture some correlations between linked nodes, but also ignore some. The strong correlations between all linked nodes can be reflected in fused embeddings based on the devised attention mechanism.

---

> ### Author Response · Authors · 2023-11-14
> **Part 6 for Reviewer u41A**
>
> **W5&Q3:**
>
> Consistent with baselines, when there are not enough |N| neighbors, only available neighbors are used for information aggregation. When a node does not have enough temporal neighbors, it indicates that there are few events currently associated with this node, which is also an evolution pattern that needs to be considered without additional operations.
>
> The purpose of setting |N|is to utilize only the most recently interacted neighbors, otherwise all neighbors in the history will be considered. Therfore, not restricting the number of neighbors may lead to an excessive number of neighbors and excessive computation, which is more serious for dense datasets like Flights and Reddit. In addition, not limiting the number of neighbors may lead to unstable training.
>
> **Weakness about typos:** Thanks again for your comments and the typos you pointed out. We have corrected them and further improved our paper. We hope our answers address your concerns and can improve your view of our work.

---

### Official Review · Reviewer_farU · 2023-10-31

**Soundness:** 3 good
**Presentation:** 3 good
**Contribution:** 3 good
**Rating:** 6
**Confidence:** 3

**Summary:**

This paper propose GLEN, an adventurous method for effective and efficient temporal graph representation learning. GLEN can generates embeddings for graph nodes by considering both global and local perspectives. Sufficient experimental results demonstrate that GLEN outperforms other baselines in both link prediction and dynamic node classification tasks.

**Strengths:**

1.	The paper clearly explain the motivation of the idea combining global and local perspectives in temporal graph representation learning, and the reason to use RNN-TCN and TGN correspondingly.
2.	The experiments in this paper are comprehensive, and they provide ample evidence of the model's superiority in terms of performance and efficiency.

**Weaknesses:**

1.	In the specific components of the model, many aspects are not novel. For example, RNN-TCN and TGN are both derived from previous works. In Cross-Perspective Fusion Module, this module is a common transformer.

2.	I still have some doubts regarding the use of RNN-TCN for extracting global information. Both GCN and TGN employ similar information aggregation approaches, aggregating nodes up to n-hops away. Why is RNN-TCN considered to be more effective in representing global information?

**Questions:**

See “Weaknesses”

---

> ### Author Response · Authors · 2023-11-14
>
> Thanks a lot for your very considerate feedback, which is very helpful for us to further improve this paper. We hope our following answers will address the points you have raised and improve your view of our work.
>
> **W1:**
>
> Thanks for your comments. In fact, we don't use any RNN in our method. As mentioned in our paper, RNNs generally suffer from inefficiency and unstable training. To avoid the problems, we innovatively adopt TCN to model the sequential effect across snapshots. Our fusion module is also not a transfomer, but a designed attention mechanism.
>
> To the best of our knowledge, we are the first in the field of temporal graph learning to propose a method that simultaneously models the graph structure from an entire global perspective and a local subgraph perspective, and fuses all node embeddings across views. Our method fills the gap in existing methods that only focus on one perspective and highlights the importance of considering both views. Our concept and related analysis of the idea that modeling temporal graphs from both global and local perspectives is novel and has research significance in this field. And we propose a completely new model based on this motivation.
>
> Therefore, our novelty lies in our contributed insights, motivation as well as the innovatively proposed method. The novelty of each module is specified below:
>
> - The novelty of the global embedding module is that there is no work that combines GNN and TCN for learning on temporal graphs to obtain more efficient and effective results, to the best of our knowledge.
> - In the local embedding module, we designed a new time interval weighting algorithm that is more concise and efficient.
> - The Cross-Perspective Fusion Module is designed to better fuse global and local embeddings, and its novelty lies in the motivation for integration based on the semantic correlation among all node embeddings, while the attention mechanism is just a way for us to implement the module.
>
> It is also of research significance to construct effective and efficient methods that work well in experiments and applications using basic, simple components. This inspires us that it may not be necessary to adopt a complex approach to temporal graph learning. Often basic models with novel and solid insights can also make good gains and achieve a better balance between inference precision and efficiency.
>
> We hope our answers address your concerns.
>
>
>
> **W2:**
>
> Thank you for your insightful suggestion.
>
> Global and local are defined according to the completeness of the graph structure being processed. Global modeling means that a snapshot of the entire graph is fed into the model when learning node representations. Graph learning operations are performed on the complete graph structure, which is done without filtering the neighbors of each node at the current time slice. Local modeling refers to processing each event of the event stream as the object and aggregating neighborhood information for the nodes involved in the events. In addition, the two perspectives model temporal patterns differently, and more elaboration can be found specifically in our introduction.
>
> Although both of these approaches aggregate information for each node within a certain range of neighborhood, TGN focuses on message passing for each node involved in interactions and use a time encoding function to fuse temporal information. This approach is more like fine-grained event-driven learning. GCN-TCN, on the other hand, captures topological information at each time slice and models sequential information across time, which is more like exploring the evolution of the graph through the model itself. Therefore, there is still some difference in the neighbor information aggregated in the two ways. As we mentioned in Appendix A, the information aggregated in the two ways is indeed going to be different. Global and local embeddings can capture some correlations between linked nodes, but also ignore some. The strong correlations between all linked nodes can be reflected in fused embeddings.
>
> In addition, our cross-perspective fusion module can break through the n-hop range restriction since it takes into account the embeddings of all nodes in the graph. We hope our answers can address your concerns.

---

> > ### Comment · Reviewer_farU · 2023-11-21
> >
> > Thanks for the authors‘ reply, which resolved most of my questions. I would like to maintain my rating.

---

> > > ### Author Response · Authors · 2023-11-22
> > >
> > > We are glad that our replies have addressed most of your concerns. Thanks again for your valuable review, which helped us to improve our paper considerably!

---

> ### Comment · Reviewer_K85V · 2023-11-22
>
> Thank you for your responses. After reading all the comments, I'd like to keep my rating.

---

> > ### Author Response · Authors · 2023-11-22
> >
> > Thanks a lot for your considerate review and positive rating. Your suggestions have helped us to further improve our work.

---

### Official Review · Reviewer_K85V · 2023-10-31

**Soundness:** 3 good
**Presentation:** 3 good
**Contribution:** 3 good
**Rating:** 6
**Confidence:** 3

**Summary:**

This paper introduces a novel Global and Local embedding Network (GLEN) for temporal graph representation learning, which captures both local and global information. Specifically, GLEN first generates both local and global embeddings, and then combine these embeddings via cross-perspective fusion module. The proposed GLEN is evaluated on several real-world datasets.

**Strengths:**

1. The idea of local embedding and global embedding is novel and interesting. For a given window of the temporal graph, GCN is used to extract node embeddings for each time stamp. TCN is used to capture global node embeddings, and a temporal interval weighting module is used over a restricted neighborhood to capture the local embeddings. In the end, the local and global embeddings are combined via a cross-perspective fusion module.
2. The proposed method could significantly outperform SOTA temporal graph embedding methods on several benchmark datasets, and the ablation study demonstrates that each of the proposed component is crucial for model's performance.
3. The writing of the paper is clear in general.

**Weaknesses:**

1. The motivation of the cross-perspective fusion module needs further clarification. Why do you use the global embedding as the query but not the local embedding? Why not (1) use global embedding as query to obtain z1, (2) use local embedding as query to obtain z2 and (3) combine z1 and z2?
2. What will happen if you only use the local embedding module but without restricting the size of neighbors?
3. Some details need further improvement.
(1). What are $\hat{y}_0, \hat{y}_1,\dots$? (Between Eq. (8) and Eq. (9))
(2). $\mathbf{h}_v^{(0)} = \mathbf{s}_v^{(t)}+\mathbf{x}_v^{(t)}$?

**Questions:**

Please refer to weaknesses.

---

> ### Author Response · Authors · 2023-11-14
> **Part 1 for Reviewer K85V**
>
> Thanks a lot for your insightful comments and detailed suggestions, which are very helpful for us to further improve this paper. We hope our following answers will address the points you have raised and improve your view of our work.
>
> **W1:**
>
> Thanks for pointing this out. Due to space constraints, we did not explain this in the paper. We use the global embedding as the query and the local embedding as the key and value for two main reasons:
>
> - In terms of modeling temporal information, local embeddings are obtained by encoding specific timestamp information, whereas global module relies more on evolution patterns across time slices by modeling sequencial information at a coarse-grained level using TCN. Therefore, in order to better utilize the fine-grained timestamp information, the hidden state of local embeddings need to be utilized.
>
> - In terms of topological information, although both local and global modules aggregate neighborhood information for nodes, the local module can aggregate neighbor information more effectively than the graph convolution of the global module due to the use of our devised  weighted sum algorithm based on time interval. And as we mentioned in the Introduction, local and global are oriented to graph structures with different degrees of integrity (local subgraphs or entire graph snapshots), so the operations of the local module are more fine-grained.
>
> To better validate the soundness of our considerations and address your concern, we further compare the performance (Average Precision, %)  of the model before and after (denoted as GLEN-swap) swapping $Z^{Global}$ and $Z^{Local}$ in the attention, as shown in the following tables:
>
> | Dataset                      |           | Wikipedia  | Wikipedia      | Wikipedia     | Reddit     | Reddit         | Reddit        | Enron      | Enron          | Enron         | UCI        | UCI            | UCI           |
> | ---------------------------- | --------- | ---------- | -------------- | ------------- | ---------- | -------------- | ------------- | ---------- | -------------- | ------------- | ---------- | -------------- | ------------- |
> | **NS Strategy**              |           | **Random** | **Historical** | **Inductive** | **Random** | **Historical** | **Inductive** | **Random** | **Historical** | **Inductive** | **Random** | **Historical** | **Inductive** |
> | Transductive link prediction | GLEN      | 99.99±0.01 | 97.67±0.16     | 95.20±0.36    | 99.99±0.01 | 98.74±1.25     | 99.24±0.82    | 93.67±0.48 | 94.35±2.24     | 94.99±1.91    | 99.55±0.90 | 98.24±1.13     | 93.30±3.31    |
> | Transductive link prediction | GLEN-swap | 96.79±0.69 | 89.97±3.45     | 84.38±11.43   | 98.19±0.10 | 86.60±2.23     | 85.13±6.27    | 88.46±0.84 | 77.79±6.31     | 71.76±2.56    | 82.56±5.08 | 77.36±1.13     | 76.80±0.62    |
> |                              | Reduction | ⬇3.2%      | ⬇7.88%         | ⬇11.37%       | ⬇1.8%      | ⬇12.29%        | ⬇14.20%       | ⬇4.86%     | ⬇17.55%        | ⬇24.46%       | ⬇17.07%    | ⬇31.43%        | ⬇28.4%        |
> | Inductive link prediction    | GLEN      | 99.95±0.05 | 96.25±0.27     | 96.13±0.29    | 99.85±0.28 | 97.31±2.46     | 97.28±2.48    | 96.15±1.61 | 97.28±0.53     | 97.38±0.46    | 99.11±0.30 | 95.94±2.00     | 95.43±2.63    |
> | Inductive link prediction    | GLEN-swap | 96.82±0.46 | 85.72±11.98    | 85.86±11.81   | 96.18±0.17 | 79.84±1.07     | 81.81±2.90    | 88.48±2.68 | 62.86±3.14     | 63.52±3.14    | 89.91±1.99 | 78.98±0.83     | 79.03±0.65    |
> |                              | Reduction | ⬇3.13%     | ⬇10.94%        | ⬇10.68%       | ⬇3.68%     | ⬇17.93%        | ⬇15.9%        | ⬇18.38%    | ⬇35.38%        | ⬇34.77%       | ⬇19.37%    | ⬇28.1%         | ⬇27.66%       |

---

> ### Author Response · Authors · 2023-11-14
> **Part 2 for Reviewer K85V**
>
> | Dataset                      |           | UN Trade    | UN Trade       | UN Trade      | MOOC       | MOOC           | MOOC          | Flights    | Flights        | Flights       |
> | ---------------------------- | --------- | ----------- | -------------- | ------------- | ---------- | -------------- | ------------- | ---------- | -------------- | ------------- |
> | **NS Strategy**              |           | **Random**  | **Historical** | **Inductive** | **Random** | **Historical** | **Inductive** | **Random** | **Historical** | **Inductive** |
> | Transductive link prediction | GLEN      | 97.83±0.15  | 97.65±0.08     | 97.45±2.11    | 98.39±2.52 | 99.89±0.11     | 99.86±0.19    | 99.96±0.01 | 85.46±0.39     | 89.83±0.77    |
> | Transductive link prediction | GLEN-swap | 83.43±11.32 | 72.62±10.57    | 72.72±10.48   | 97.87±3.92 | 85.26±3.49     | 88.27±5.24    | 78.88±3.92 | 79.92±5.33     | 79.61±14.67   |
> |                               | Reduction | ⬇14.72%     | ⬇25.63%        | ⬇25.38%       | ⬇0.53%     | ⬇14.65%        | ⬇11.61%       | ⬇21.09%    | ⬇6.48%         | ⬇11.49%       |
> | Inductive link prediction    | GLEN      | 96.09±0.12  | 95.78±2.32     | 95.76±2.32    | 96.48±4.02 | 99.53±0.93     | 99.54±0.91    | 99.36±0.17 | 76.96±0.54     | 77.23±0.61    |
> | Inductive link prediction    | GLEN-swap | 80.16±8.71  | 69.59±7.56     | 69.58±7.51    | 95.50±3.54 | 85.98±3.84     | 85.88±3.69    | 77.50±3.54 | 75.43±4.39     | 75.58±5.25    |
> |                               | Reduction | ⬇16.58%     | ⬇27.34%        | ⬇27.34%       | ⬇1.02%     | ⬇13.61%        | ⬇13.72%       | ⬇22%       | ⬇1.99%         | ⬇2.14%        |
>
> The performance decreases on all datasets after swapping $Z^{Global}$ and $Z^{Local}$ . The experimental results somewhat justify our setting of the query, key and value.

---

> ### Author Response · Authors · 2023-11-14
> **Part 3 for Reviewer K85V**
>
> **W2:**
>
> Thanks for catching this. Limiting the number of temporal neighbors is to utilize only the most recently interacted neighbors, otherwise all neighbors in the history will be considered. Therfore, not restricting the number of neighbors may lead to an **excessive number of neighbors and excessive computation**, which is more serious for dense datasets like Flights and Reddit. In addition, not limiting the number of neighbors may lead to **unstable training**.
>
> Existing works on temporal graphs all have imposed a limit on the neighborhood size, so we do so as well. The main reasons for doing so are summarized as follows:
>
> - To simplify the problem and fix the computation pattern. To avoid too many neighbors of some nodes leading to excessive computational overhead and time cost. A large amount of existing works demonstrate that limiting the number of neighbors can strike a good balance between accuracy and efficiency.
> - For the sake of stable training. As mentioned in the paper TGAT [1], sampling from neighborhood, or known as neighborhood dropout, may speed up and stabilize model training. For temporal graphs, neighborhood dropout can be carried uniformly or weighted by the inverse timespan such that more recent interactions has higher probability of being sampled.
> - For the sake of comparison fairness. All baselines limit the neighborhood size and investigate it as a hyperparameter. In order to allow a fair comparison between our method and baselines, it is inappropriate to change this operation.
>
> In addition, the global module processes the graph structure at each time slice. If only use the local embedding module but without restricting the size of neighbors, all interactions in the history are considered by the local module, which is inconsistent with the role of the global embedding module. The concept of "global" emphasizes the integrity of the graph structure at each time slice rather than considering the full historical information.
>
> **W3:**
>
> (1)The sequence of $\hat{y}$ is the output sequence after the TCN performs the causal convolution operation on the input sequence $\{x_0,x_1,...\}$. As drawn in Figure 2, the TCN outputs an output sequence with the same length as the input sequence. For $d$-dimensional eigenvectors, we consider the values at different timestamps for each dimension as a time series. So the convolution operation is performed in $d$ channels, each using the last output element $\hat{y}_{\Gamma-1}$ as the predictive value.
>
> (2)What you have written may refer to the sentence "$h^{(0)}_v(t)$ is the sum of $s_v(t)$ and temporal node features." Differently from the  original baseline TGAT [1] where no node-wise temporal features were used, in our local module the input representation of each node $v$ is $h^{(0)}_v(t)=s_v(t)+x_v(t)$ and as such it allows the model to exploit both the current memory $s_v(t)$ and the temporal node features $x_v(t)$.
>
> Thanks again for your advice. Due to space constraints, some of our statements may not be clear. We will consider adding relevant explanations to the paper to make it clearer.
>
> [1] Da Xu, Chuanwei Ruan, Evren Korpeoglu, Sushant Kumar, and Kannan Achan. Inductive representation learning on temporal graphs. In ICLR, 2020. URL https://openreview.net/forum?id=rJeW1yHYwH.

---

### Official Review · Reviewer_L1Kz · 2023-11-01

**Soundness:** 3 good
**Presentation:** 3 good
**Contribution:** 2 fair
**Rating:** 5
**Confidence:** 4

**Summary:**

The paper discusses the limitations of existing methods in temporal graph learning, which either focus on global or local perspectives but not both. To overcome this, the Global and Local Embedding Network (GLEN) is proposed. GLEN dynamically generates node embeddings by considering both global and local information. These embeddings are then fused using a cross-perspective module to capture high-order semantic relations. GLEN has been evaluated on multiple datasets and outperforms baselines in tasks like link prediction and dynamic node classification.

**Strengths:**

S1. Temporal graph is an important problem to address in practical world, yet majority of the research deals with static graphs only.

S2. The analysis of existing RNN-based work and random walk/message passing models provides useful insights.

S3. The writing/organization of the paper is generally clear, although some parts need more clarification. (See W2)

**Weaknesses:**

W1. The proposed model, while technically valid and sound, is not sufficiently novel or exciting. Combining local and global perspectives are common ideas in graphs. Even on temporal graph, point process based modeling aims to capture the graph-wide evolution pattern from  a global perspective, such as (Lu et al., 2019) and the below paper [a]. A detailed discussion on temporal point processes for temporal graph is warranted, potentially with additional experimental comparison.

[a] Trend: Temporal event and node dynamics for graph representation learning. WWW 2022.

W2. Certain parts in the motivation of the paper are not clearly explained. For example, the following sentences:
"Pairwise interactions observed in different graphs or even the same temporal graph typically have different temporal properties."
"Since the endogenous and exogenous factors driving the generative process ..."
I'm not exactly sure how they directly connect to or motivate the proposed method.

W3. In Table 2, Random tends to perform the best compared to historical/inductive strategies. It is surprising and more discussion is needed. (Also, I'm not confident of the results in Table 2, as it has some discrepancy with the results in Table 3 -- e.g. for UCI, the results in Table 2 and Table 3 are different.

**Questions:**

Please see Weaknesses.

---

> ### Author Response · Authors · 2023-11-14
> **Part 1 for Reviewer L1Kz**
>
> Thanks a lot for your insightful comments and detailed suggestions, which are very helpful for us to further improve this paper. We hope our following answers will address the points you have raised and improve your view of our work.
>
> **W1:**
>
> Thank you for your valuable comments. We provide relevant discussions about the two methods using temporal point processes as follows. We will add relevant discussions of these methods in the future versions of our paper.
>
> **Discussions about the paper (Lu et al., 2019):**
>
> The method for macro- and micro-dynamics proposed in the paper (Lu et al., 2019) are quite different from our devised modules of global and local perspectives.
>
> - For micro-dynamics, they "regard the establishments of edges as the occurrences of chronological events and propose a temporal attention point process to capture the formation process of network structures". Therefore, the temporal point process they use doesn't aims to capture the graph-wide evolution pattern from a global perspective. Time point processes are often based on assumptions such as events occurring as a random process based on conditional intensity, and the effect of historical events on the intensity function is cumulative. These assumptions are not generally applicable to temporal graphs. Our devised local embedding module based on time interval weighting fully considers the temporal nature to aggregate information about neighbors and events more effectively.
> - For macro-dynamics, they use "a general dynamics equation parameterized with network embeddings". While our global perspective modeling uses graph neural networks and TCN, which are better suited for modeling the graph structures and sequential information.
> - In addition, they set the loss function terms for macro and micro dynamics respectively, which makes predictions more dependent on the model's capability itself. Whereas, our method fuses global and local embeddings through a devised attention mechanism, which helps adaptively capture the semantic correlations between global embeddings and local embeddings of all nodes.
>
> **Discussions about TREND:**
>
> The primary motivation of TREND is to explicitly capture the exciting effects between events through the temporal point processes, most notably the Hawkes process. A Hawkes process is a stochastic process that can be understood as counting the number of events up to time 𝑡. Its behavior is typically modeled by a conditional intensity function 𝜆(𝑡), the rate of event occurring at time 𝑡 given the past events. The paper mentions: "integrating the node dynamics provides a regularizing mechanism beyond individual events, to ensure that **the events from a node, as a collection**, conform to the continuous evolution of the node." Thus, methods like TREND focus mainly on processing the collection of events about each node.
>
> Here "collection" corresponds to the set of events related with temporal neighbors of a single node, which is different from our global perspective that processes the entire graph structure. This operation is more like the local perspective we defined, since it is oriented to a local subgraph of each node. Our global module focuses on graph representation learning on the complete graph structure at each time slice, which has **a larger scope compared to approaches that model the event collection of each node**.
>
> To indicate the difference between our method and methods using temporal point processes, we show here the inductive link prediction performance (in percent, with the random negative sampling strategy) comparison between our approach GLEN and TREND, as TREND is the SOTA method that uses the temporal point processes:
>
> | Dataset | Wikipedia  | Wikipedia  |   Reddit   |   Reddit   |    UCI     |    UCI     |   Enron    |   Enron    |  UN Trade  |  UN Trade  |    MOOC    |    MOOC    |  Flights   |  Flights   |
> | ------- | :--------: | :--------: | :--------: | :--------: | :--------: | :--------: | :--------: | :--------: | :--------: | :--------: | :--------: | :--------: | :--------: | :--------: |
> |         |  Accuracy  |     F1     |  Accuracy  |     F1     |  Accuracy  |     F1     |  Accuracy  |     F1     |  Accuracy  |     F1     |  Accuracy  |     F1     |  Accuracy  |     F1     |
> | TREND   | 83.75±1.19 | 83.86±1.24 | 83.06±0.24 | 83.65±0.27 | 65.05±0.92 | 63.15±2.54 | 73.58±2.49 | 69.79±2.42 | 75.40±2.11 | 74.18±1.65 | 76.36±2.55 | 79.20±3.40 | 73.26±0.02 | 73.52±0.32 |
> | GLEN    | 97.44±0.97 | 96.14±1.04 | 98.26±0.21 | 96.86±0.39 | 76.80±5.52 | 82.03±1.60 | 84.02±3.29 | 85.55±1.62 | 89.13±1.55 | 91.96±1.09 | 87.77±3.09 | 91.13±1.74 | 85.32±0.75 | 85.65±1.78 |
> | Improv. |  ⬆16.35%   |  ⬆14.64%   |  ⬆18.31%   |  ⬆15.79%   |  ⬆18.06%   |  ⬆23.02%   |  ⬆14.19%   |  ⬆22.58%   |  ⬆18.21%   |  ⬆23.97%   |  ⬆14.94%   |  ⬆15.06%   |  ⬆16.46%   |  ⬆16.50%   |

---

> ### Author Response · Authors · 2023-11-14
> **Part 2 for Reviewer L1Kz**
>
> The experimental results above somewhat demonstrate the advantages of our approach over methods that use temporal point processes. In addition, the training efficiency of our model is much higher than TREND. Due to the high dependence on the number of historical events and node features, methods using temporal point processes may perform poorly on sparse or absent node features datasets (such as UCI and Enron).
>
> **W2:**
>
> Thanks for catching this.
>
> These sentences are elaborated mainly to **explain our motivation for introducing information from both global and local perspectives**.
>
> - The first sentence indicates that graph topologies across domains or across time may have quite different temporal properties. For example, it is clear that the evolution patterns of the temporal graphs in the economics (UN Trade), social interaction (Reddit) and transport (Flights) domains are quite different. Due to the regularity and abruptness of events, the pattern of events can vary across time. Therefore, modeling at different time granularities have to be taken into account, implying the necessity for us to utilize both global and local modules.
> - The second sentence demonstrates the diversity of graph evolution. The complexity of endogenous and exogenous factors that drive changes in graph structure can make temporal graphs both regular and mutant. Thus this also indicates the need to introduce both global and local perspective modeling for adaptively capturing different graph natures.
>
> We will consider revising the statement here to make it clearer in the future version of our paper. We hope our answers address your concerns.
>
> **W3:**
>
> Thanks a lot for pointing this out. We did have a data recording error in Table 3. The actual data about the UCI dataset should be consistent with Table 2 and we have corrected it in the paper.
>
> The historical and inductive negative sampling strategies are first proposed in this benchmark paper [1]. Before this benchmark was proposed, the evaluation of link prediction for temporal graph methods was evaluated only using the random negative sampling strategy. **This benchmark paper points out the following problems with the random strategy :**
>
> - No Collision Checking: It is possible for the same edge to be both positive and negative. This collision is more likely to happen in denser datasets, such as UN Trade.
> - No Reoccurring Edges: The probability of sampling an edge which was observed before is often very low due to the sparsity of the graph. However, in many real-world tasks such as flight prediction, correct prediction of the same edge for different time steps is particularly important.
>
> Thus, the random strategy may exaggerate the efficacy of current models on real-world tasks and hinder researchers’ ability to evaluate if new models are superior. **Historical and Inductive strategies are more stringent and challenging evaluation procedures for link prediction specific to temporal graphs, which reflect real-world considerations** and better compare the strengths and weaknesses of methods. We introduced these negative strategies in Appendix 4.5 and highlight them here for your convenience:
>
> - Historical Negative Sampling: Historical negative sampling strategy samples negative edges from the set of edges that have been observed during previous timestamps but are absent in the current step in order to evaluate whether a given method is able to correctly predict an observed training edge would reoccur.
> - Inductive Negative Sampling: Unlike historical negative sampling, the objective of inductive negative sampling is to evaluate whether a given method can model the reoccurrence pattern of edges only seen during test time. Thus, this strategy samples negative edges from the test instances that were not observed during training and are also absent currently.
>
> **The benchmark paper also points out that existing methods have significant performance degradation with the evaluation of both historical and inductive strategies.** Therefore, it's normal that random tends to perform the best compared to historical/inductive strategies.
>
> [1] Poursafaei, Farimah , et al. "Towards Better Evaluation for Dynamic Link Prediction." (NIPS, 2022). URL https://openreview.net/forum?id=1GVpwr2Tfdg.

---

> > ### Comment · Reviewer_L1Kz · 2023-11-18
> >
> > Thanks for the detailed responses. I will weigh them carefully.

---

> ### Author Response · Authors · 2023-11-18
>
> Thanks a lot for your response and discretion. We welcome any other questions and further discussions with us. We sincerely hope that our replies can improve your perception of our work.

---

> ### Comment · Reviewer_L1Kz · 2023-12-03
>
> Based on the response, I would increase my score to 5. Thanks.

---

### Public Comment · ~Ruixing_Zhang1 · 2023-11-20
**A doubt of a statement in the paper**

In your paper, the author stated that To the best of their knowledge, they are the first in the field of temporal graph learning to propose a method that simultaneously models the graph structure from an entire global perspective and a local subgraph perspective. Actually, a similar idea is presented in [1] which uses hierarchical memories to recept fine-grained and macro-grained information. Briefly, in [1], they maintain many memories for each node, the finest one is triggered whenever the edge comes. The more macro ones will be updated when a snapshot of the graph with length $30 * 2^k $ occurs. Because they also model the traffic network as a temporal graph, I doubt the rigor of your statement.

[1] Ruixing Zhang, Liangzhe Han, Boyi Liu, Jiayuan Zeng, & Leilei Sun (2022). Dynamic Graph Learning Based on Hierarchical Memory for Origin-Destination Demand Prediction. In Proceedings of the Thirty-First International Joint Conference on Artificial Intelligence, ĲCAI 2022, Vienna, Austria, 23-29 July 2022 (pp. 2383–2389). ijcai.org.

---

> ### Author Response · Authors · 2023-11-20
>
> Thank you for raising this point. This work [1] may be only slightly similar to ours in the general direction, but it differs from ours in many ways, including the specific implementation of the model, the problems it is oriented towards, and so on. We summarize the differences below:
>
> - The fine-grained and macro-grained mentioned in this paper [1] are not quite the same as our concepts of global and local perspectives. Their method obtains fine-grained and macro-grained information in the same way. Fine-grained and macro-grained are mainly reflected in the time span of their memory updates.
>
>   Our approach devises different modeling methods and new algorithms for both global and local perspectives, which are more specific and relevant for obtaining information with different spatial and temporal granularities. Global and local are defined according to the completeness of the graph structure (the entire graph and subgraphs) being processed.
>
> - This paper's combination of fine-grained and macro-grained lies in its message fusion mechanism and the concatenation of each level’s memory. Their message fusion mechanism "is proposed to integrate the specific track message with others to integrate multiple types of spatiotemporal information." [1]
>
>   Our approach, on the other hand, fuses embeddings from both global and local perspectives. We use modeling with different perspectives to embed spatiotemporal information of different granularities into eigenvectors for attentional fusion, rather than fusing messages beforehand or simply concatenating memories before generating embeddings. Our fusion module considers the attentional relevance of the eigenvectors of all nodes, rather than only considering information from the same node at different spatio-temporal granularities.
>
> - Compared to this paper [1], our work is more versatile and generalized, oriented to a broader domain, and our experiments are more comprehensive. This paper focuses the the prediction of origin-destination (OD) demands, and the experiments were conducted on only 2 datasets. This paper is more like an application of dynamic graph representation learning to the OD domain and is not compared to state-of-the-art methods.
>
>   Whereas our work is more generalized to temporal graph representation learning. As mentioned in our paper, we used multiple datasets of different scales in various domains such as social, transport, economics, and so on. Additionally, we evaluated our method on multiple prediction tasks and with more evaluation strategies.
>
> - The memory updater in this paper [1] relies on GRUs, which may bring about the problems of inefficiency and unstable training as we mentioned in our paper.
>
>   Our approach innovatively uses TCNs instead of RNNs in the global embedding module to provide higher training efficiency and stability. To the best of our knowledge, we are the first to combine GNN with TCN in the temporal graph learning subfield.
>
> - In terms of efficiency, we use experiments and analysis to validate the efficiency advancement of our approach, which is lacking in this work [1].
>
> - In terms of motivation, this paper's motivation for the synthesis of fine-grained and macro-grained information is not fully validated, whereas we validate the soundness of our consideration of both global and local perspectives in Appendix A.
>
> We appreciate you discussing this paper of yours with us. Since this paper is not oriented towards pure temporal graph representation learning and is not very relevant to our paper, we did not take it into account before. We will consider researching more work in the future. Thanks again for your comments.
>
> [1] Ruixing Zhang, Liangzhe Han, Boyi Liu, Jiayuan Zeng, & Leilei Sun (2022). Dynamic Graph Learning Based on Hierarchical Memory for Origin-Destination Demand Prediction. In Proceedings of the Thirty-First International Joint Conference on Artificial Intelligence, ĲCAI 2022, Vienna, Austria, 23-29 July 2022 (pp. 2383–2389). ijcai.org.

---

> > ### Public Comment · ~Ruixing_Zhang1 · 2023-11-22
> >
> > Thank you for your detailed comment. Besides, combining TCN and GNN may have been proposed before 2019 in spatiotemporal data mining[1]. Since there is no further modification, the contribution of the first to combine GNN with TCN in the temporal graph learning subfield is still doubtable.
> >
> > [1] Wu, Zonghan, et al. "Graph wavenet for deep spatial-temporal graph modeling." Proceedings of the 28th International Joint Conference on Artificial Intelligence. 2019.

---

> ### Author Response · Authors · 2023-11-22
>
> Thank you for your comment. We also read this when we were constructing our work. Our statements on this point are as follows:
>
> - Spatio-temporal graphs have certain differences from temporal graph data. Our work and this work are oriented towards different problem settings. The goal of spatio-temporal graph modeling in this work [1] is to predict node attributes, which is a regression problem. Such methods aim at modeling historical graph signals, while our work and relevant baselines focus on link prediction and node classification. Actually, for the field of spatio-temporal graphs, there is an earlier work that also proposes to combine GCN with TCN [2], which is oriented towards the skeleton-based action recognition.
>
>   Therefore, we only said that our work is the first "in the temporal graph learning subfield", instead of the first "in spatiotemporal data mining". You can see that the baselines most relevant to our work all use conventional RNNs rather than TCN. Adopting the methodology for different tasks and domains is also research-worthy.
>
> - As we stated in the paper and our replies to reviewers, the most innovative point and motivation of our work is the insight of modeling from both global and local perpectives and the effective fusion of features. We designed specific modules for each of the two perspectives, where the combination of GCN and TCN is just the way we implemented the global module.
>
> We thank again for your concern and advice on our work.
>
> [1]Wu, Zonghan, et al. "Graph wavenet for deep spatial-temporal graph modeling." Proceedings of the 28th International Joint Conference on Artificial Intelligence. 2019.
>
> [2]Yan, Sijie, Yuanjun Xiong, and Dahua Lin. "Spatial temporal graph convolutional networks for skeleton-based action recognition."*Proceedings of the AAAI conference on artificial intelligence*. Vol. 32. No. 1. 2018.

---

> > ### Public Comment · ~Ruixing_Zhang1 · 2023-11-24
> >
> > Thank you. And now I know this work's contribution.

---

### Author Response · Authors · 2023-11-21

Dear Reviewers,
We sincerely appreciate your valuable review. In response to your constructive comments, we have:
(1) conducted additional experiments (including two new baselines and more ablation experiments);
(2) provided more analysis of other baselines and our method;
and (3) carefully prepared our response to your questions.
We hope that the additional results could address your concerns and enhance the clarity of our problem settings, observations, motivations, and methods. As the author response period is coming to an end, we would appreciate it if you could consider our response and we are more than willing to address any further comments or questions.
Once again, we thank the reviewers for the valuable feedback, which has undoubtedly contributed to improving the quality of our work.

---

### Meta-Review · Area_Chair_93dt · 2023-12-13

**Metareview:**

The paper studies temporal graph representation learning. It presents the Global and Local Embedding Network (GLEN), which is argued to address the shortcomings of current temporal graph learning methods that either concentrate on global or local aspects but not both. GLEN produces node embeddings by integrating global and local data, and then combines these embeddings through a cross-perspective module to grasp complex semantic relations. The experimental results suggest that GLEN outperforms benchmarks in link prediction and dynamic node classification.

The studied problem is of importance to graph learning and mining. The reviewers also find that the presentation of the paper is generally clear, although some parts need more clarification.

The major concern is on the technical novelty of this work. The claimed technical contribution mostly lies in the combination of local and global information for learning embeddings for (temporal) graphs. Several reviewers question about this. In fact, this has been widely studied and explored in existing graph learning literature as pointed out by the reviewers (and public comments).

In addition, the experiments are performed on very small graph datasets. Considering the availability of larger, dynamic datasets, like those in ROLAND [KDD'22], a more compelling empirical demonstration is possible. In addition, the reviewers have identified inconsistencies in accuracy results across different tables, highlighting the need for thorough verification and check before publishing.

**Justification For Why Not Higher Score:**

Though the studied problem -- temporal graph representation learning --- is of importance, the proposed method and experimental results revealed do not offer exciting technical perspective or insights into this problem. The reviewers also rate it just below the threshold (6, 6, 5, 5) following the rebuttal. Essentially, it does not contain a compelling technique or result that attract anyone involved.

**Justification For Why Not Lower Score:**

n/a

---

### Decision · Program_Chairs · 2024-01-16

Reject